# Coordination modulation of hydrated zinc ions to enhance redox reversibility of zinc batteries

Song Chen [1], Deluo Ji[1], Qianwu Chen[1], Jizhen Ma[1], Shaoqi Hou[2] & Jintao Zhang [1] ✉

The dendrite growth of zinc and the side reactions including hydrogen evolution often degrade performances of zinc-based batteries. These issues are closely related to the desolvation process of hydrated zinc ions. Here we show that the efficient regulation on the solvation structure and chemical properties of hydrated zinc ions can be achieved by adjusting the coordination microenvironment with zinc phenolsulfonate and tetrabutylammonium 4-toluenesulfonate as a family of electrolytes. The theoretical understanding and in-situ spectroscopy analysis revealed that the favorable coordination of conjugated anions involved in hydrogn bond network minimizes the activate water molecules of hydrated zinc ion, thus improving the zinc/electrolyte interface stability to suppress the dendrite growth and side reactions. With the reversibly cycling of zinc electrode over 2000 h with a low overpotential of 17.7 mV, the full battery with polyaniline cathode demonstrated the impressive cycling stability for 10000 cycles. This work provides inspiring fundamental principles to design advanced electrolytes under the dual contributions of solvation modulation and interface regulation for high-performing zinc-based batteries and others.

Zn metal batteries have attracted continuous attention with advantages of low cost, environmental-friendly, high specific capacity (820 mAh g$^{-1}$ and 5855 mAh cm$^{-3}$) and good safety[1]. However, the Zn dendrite growth, side-reactions with hydrogen evolution and surface passivation would degrade the battery performance and cycling stability seriously[2]. Many strategies have been developed to stabilize Zn anode and extend its life-span, such as the coating of protective layers, the introduction of electrolyte additives[3–5]. With the significant effect to regulate the interfacial micro-environment and the solvation status of Zn$^{2+}$ ions[6–8], the easy formation of passivation layer would also lower the coulombic efficiency (CE)[9–11]. Essentially, the formation of solvation structure, $[Zn(H_2O)_6]^{2+}$ would result in the high energy barrier of desolvation, thus inhibiting the smooth zinc deposition. Increasing electrolyte concentration could be a feasible method to decrease the active water molecules[12]. Thus, "water-in-salt" electrolytes (e.g., 1 M Zn(TFSI)$_2$ and 20 M LiTFSI) were proposed to form closely Zn-TFSI$^+$ ion pairs instead of $[Zn(H_2O)_6]^{2+}$ [13]. However, the high concentration electrolyte with high cost would increase the viscosity with large ion migration resistance, thus slowing down the reaction kinetics.

Alternatively, the introduction of additives is a feasible and economic method to tackle the issues for metal deposition that has been commonly studied in the electroplating industry[14]. Typically, the additives absorbed on the protuberances would modulate the accumulated electric field to inhibit the forced diffusion of metal ions for the further dendrites growth[15]. The coverage of additives also protects Zn anode from the interfacial corrosion with hydrogen evolution. It's worth noting that the adsorption ability should be considered because the intense adsorption would cause severe polarization at electrode-

[1]Key Laboratory for Colloid and Interface Chemistry, Ministry of Education, School of Chemistry and Chemical Engineering, Shandong University, Jinan 250100, China. [2]School of Mathematical and Physical Sciences, University of Technology Sydney, Ultimo, NSW 2007, Australia. ✉e-mail: jtzhang@sdu.edu.cn

electrolyte interface. Therefore, it is urgent to optimize zinc salt electrolyte for reversible redox reactions of zinc with fast reaction kinetics, uniform deposition and high CE. The formation of inert products, $(Zn_4SO_4(OH)_6 \cdot xH_2O)$ has been revealed to inhibit the rapid transfer of $Zn^{2+}$ ions and smooth deposition due to the serious solvation with water molecules, resulting in poor cycling life[16].

Herein, we propose the zinc phenolsulfonate $(Zn(PS)_2)$ with a bulky and conjugated anion to modulate the solvation chemistry of zinc ions via the coordination process for improving rechargeable battery performance. As the common deodorant and anti-perspirant, $Zn(PS)_2$ is easily available for the large-scale applications[17]. The large anion would reduce the coordination number with water molecule. In the presence of tetrabutylammonium 4-toluenesulfonate (TBATS) with moderate adsorption ability, the preferential adsorption of quaternary ammonium cations $(TBA^+)$ is helpful to inhibit the 2D diffusion of $Zn^{2+}$ ions, thus inducing uniform Zn deposition. Typically, the optimized electrolyte of 1 M $Zn(PS)_2$ and 0.2 mg mL$^{-1}$ TBATS makes the batteries have a long cycle life of 2000 h and high coulombic efficiency (CE) of 99.1%, which is even better than those of the state-of-art Zn-Zn

symmetric cells with various electrolytes reported recently. Besides, the assembled pouch cell also demonstrates improved performance. The results provide the fundamental guidance to develop advanced electrolyte for high-performance ZIBs through solvation-shell modification and interface regulation strategy.

## Results

### Electrochemical performance and morphology characterization of Zn-Zn symmetric cells

In comparison with the short life span of around 90 h for the Zn-Zn symmetric cell in 1 M $ZnSO_4$ electrolyte, the cycle life increases to 202 h in 1 M $Zn(PS)_2$ electrolyte with the small voltage hysteresis of 16.4 mV (Supplementary Figs. 1, 2). The cycling stability in $Zn(PS)_2$ electrolyte can be further enhanced and exhibits the concentration-dependent durability of 402, 676, 2000, 1500, 1236 and 1182 h with increasing TBATS concentration from 0.05, 0.1, 0.2, 0.5, 1 to 10 mg mL$^{-1}$ (Fig. 1a). With the optimal concentration of 0.2 mg mL$^{-1}$, the Zn-Zn symmetric cell could still cycle steady for more than 1000 h even at a larger current density of 3 mA cm$^{-2}$ (Supplementary Fig. 3).

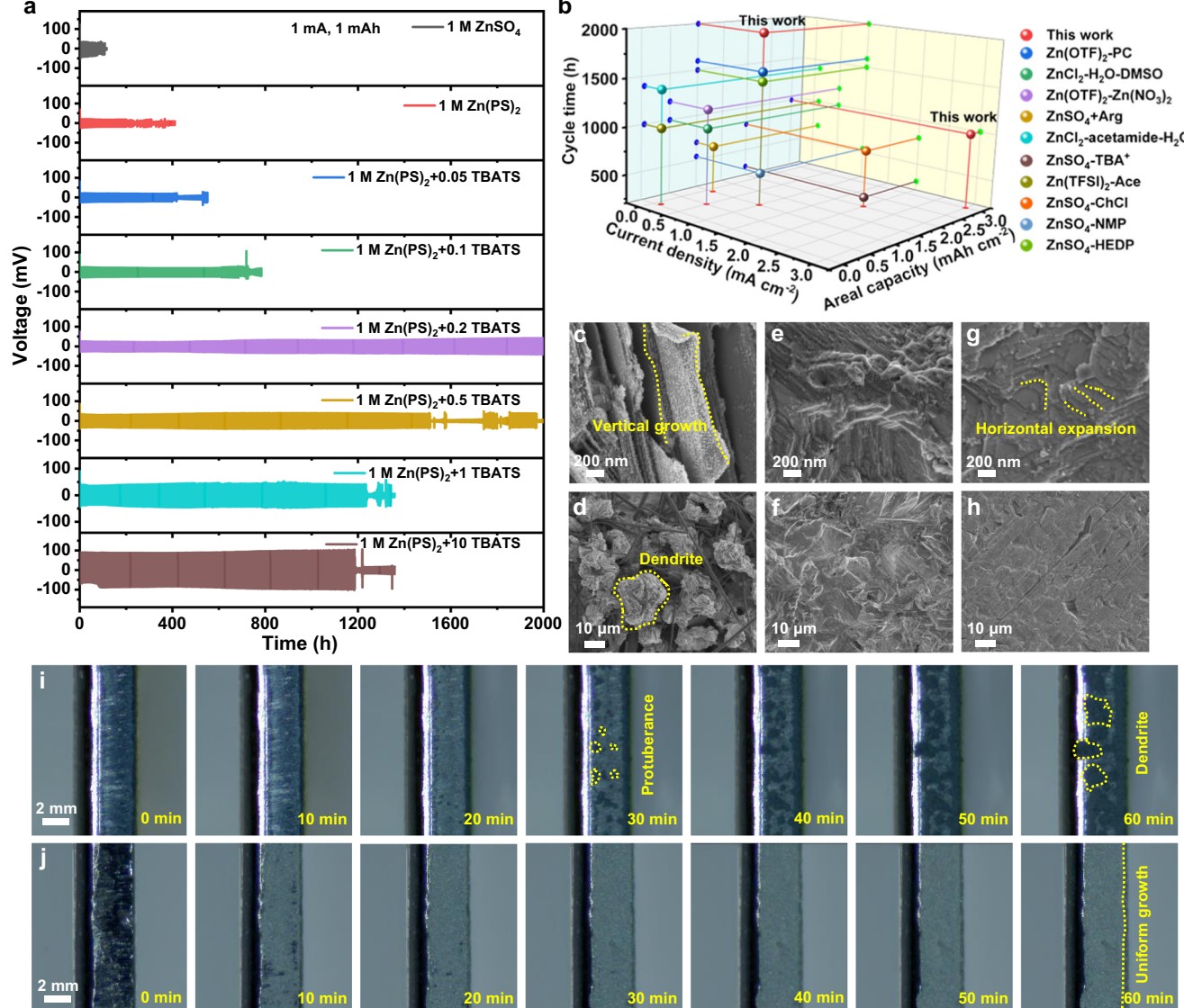

**Fig. 1 | Electrochemical and morphology characterization. a** Voltage-time curves of Zn-Zn symmetric cells at 1 mA cm$^{-2}$, 1 mAh cm$^{-2}$. **b** Comparison of cycling stability with recently reported results at different current density and capacity. SEM images of Zn foil after 100 cycles at 1 mA cm$^{-2}$, 1 mAh cm$^{-2}$ in (**c, d**) 1 M $ZnSO_4$, (**e, f**) 1 M $Zn(PS)_2$ and (**g, h**) 1 M $Zn(PS)_2$ + 0.2 TBATS electrolytes. In situ optical microscopy observations of the Zn deposition process at a current density of 5 mA cm$^{-2}$ in (**i**) 1 M $ZnSO_4$ and (**j**) 1 M $Zn(PS)_2$ + 0.2 TBATS electrolytes.

However, the excess TBA$^+$ cations would cover the electrode surface indiscriminately, resulting in the increased voltage polarization[18]. The nucleation overpotential of Zn$^{2+}$ ion in 1 M Zn(PS)$_2$ electrolyte is 44.9 mV, much lower than that in 1 M ZnSO$_4$ (91.8 mV). Notably, the presence of TBATS does not increase the polarization with the low nucleation overpotential (50.2 mV). The platform overpotential is 22.6 mV at the 1st cycle and decreases to 17.7 mV at 200th cycle, suggesting the favorable plating/striping process (Supplementary Fig. 4 and Supplementary Table 1). The lower voltage hysteresis under changing current density from 0.5 to 10 mA cm$^{-2}$ in 1 M Zn(PS)$_2$ suggest the good rate performance (Supplementary Fig. 5). Especially, the performance is even better than those of the state-of-art Zn-Zn symmetric cells with various electrolytes reported recently (Fig. 1b)[15,18–26]. The SEM images exhibit the formation of irregular zinc flakes aggregated into large dendrites in 1 M ZnSO$_4$ electrolyte. In comparison with the randomly distributed gullies on the zinc surface in 1 M Zn(PS)$_2$ electrolyte, the flat surface is maintained-well with the horizontal expansion model of Zn deposition in 1 M Zn(PS)$_2$ + 0.2 TBATS. The results revealed that the leveling effect of TBA$^+$ cation would contribute to the smooth zinc deposition (Fig. 1c-h and Supplementary Fig. 6).

From the in situ optical images (Fig. 1i), the scattered dark zinc appears on the side after the current applied for 30 min in 1 M ZnSO$_4$ electrolyte. The obvious color difference is observed on the Zn foil owing to the uneven deposition (Supplementary Fig. 7). The zinc protrusion gradually grows into the obvious zinc dendrites with the extension of deposition time. The loose deposition of Zn makes the zinc foil significantly thicker (about 21, 16 μm) (Supplementary Figs. 8, 9). In contrast, the uniform growth of Zn with the compact layer of 9.6 μm is achieved in Zn(PS)$_2$ + 0.2 TBATS, which is close to the theoretical thickness (-8.5 μm under 5 mAh cm$^{-2}$, Fig. 1j and Supplementary Fig. 8e, f). The increasing current in ZnSO$_4$ and Zn(PS)$_2$ electrolytes (Supplementary Fig. 10) indicates the continuous formation of zinc with increasing surface area, possibly resulting in the dendrites aggregation[27]. When TBATS added, the current quickly reaches a stable value after a short decrease, which would be attributed to the overlapped growth, benefitting from the preferential coverage of TBA$^+$ to inhibit the dendrite growth[15]. With a deep plating/stripping cycle at the capacity of 10 mAh cm$^{-2}$ to remove oxidant impurities on substrate, the short circuit occurred after 63 h in 1 M ZnSO$_4$ electrolyte would be contributed to the vigorous growth of dendrites (Supplementary Fig. 11a). In contrast, the coulombic efficiency in 1 M Zn(PS)$_2$ and 1 M Zn(PS)$_2$ + 0.2 TBATS electrolyte is 97.2 and 99.1% for 100 cycles, respectively (Supplementary Fig. 11b, c). The TBA$^+$ cations as leveling agent for homogenous Zn deposition would hamper the direct interaction between zinc and water molecules to suppress water decomposition, thus achieving the high CE[21]. The good stability in Zn(PS)$_2$ + 0.2 TBATS electrolyte was also proved by immersing Zn foil in various electrolytes for one week (Supplementary Figs. 12-16 and Supplementary Table 2). From the Tafel plot test, the corrosion current density in 1 M Zn(PS)$_2$ is 3.09 mA cm$^{-2}$, smaller than that in 1 M ZnSO$_4$ (6.34 mA cm$^{-2}$). With the positive potential shift, the corrosion current density further decreases to 1.48 mA cm$^{-2}$ in the presence of TBATS as shielding layer (Supplementary Fig. 17), suggesting the corrosion inhibition to hydrogen evolution.

## Electrolyte structures

In comparison with the distorted redox peaks of zinc in ZnSO$_4$, the Zn-Zn symmetric cell with Zn(PS)$_2$ electrolyte (Fig. 2a) demonstrates the enhanced redox behaviors. The slight smaller current density suggests the significant effect on the redox process of zinc in the presence of TBATS. The symmetrical redox peaks of zinc with smallest polarization in Ti-Zn asymmetric cell further demonstrate the improved reversibility in 1 M Zn(PS)$_2$ with TBATS (Fig. 2b). Moreover, lower nucleation potentials of 90 and 73 mV are presented on Ti substrate in 1 M Zn(PS)$_2$

electrolyte with/without TBATS in comparison with the nucleation potential of 101 mV in 1 M ZnSO$_4$ electrolyte (Supplementary Fig. 18). The results exhibit the unique features of such composite electrolytes for the favorable deposition of zinc. In comparison with the common form of zinc-hexaquo complex in diluted solutions (Supplementary Fig. 19), namely the solvent shared ion pair (SSIP, [Zn$^{2+}$(H$_2$O)$_6$ · anion]), one of the inner H$_2$O molecules coordinated would be replaced by anion to form a compact ion pair (CIP, [Zn$^{2+}$(H$_2$O)$_5$ · anion]) with the increasing concentration. Typically, the Raman spectra (Supplementary Fig. 20) exhibit the increasing vibration of Zn$^{2+}$-SO$_4^{2-}$ at 260 cm$^{-1}$ owing to the increased number of CIP. The intensified peaks of Zn$^{2+}$-H$_2$O vibration (390 cm$^{-1}$) are also observed due to the increasing concentration of Zn$^{2+}$ ions. The symmetric vibration of SO$_4^{2-}$ [$v_s$(SO$_4^{2-}$)] at about 980 cm$^{-1}$ is fitted into SSIP and CIP species and the peak area of CIP increases with the increased concentration. The stretching vibration peak of OH$^-$ group typically situated at 3000-3750 cm$^{-1}$ also reflects the change of coordination environment. The hydrogen bond network can be distinguished by the involved molecule with surrounding water molecules either as proton acceptor (A) or donor (D). Generally, the main hydrogen network can be expected to be DAA-OH, DDAA-OH, DA-OH, DDA-OH, and free OH. The blue shift with increasing proton donor component indicates the more stable hydrogen bond network[16,28]. The SO$_4^{2-}$ and PS$^-$ anions associated with H$_2$O molecules are proton donor that prevents H$_2$O molecules from the breakage of O-H bonds to inhibit hydrogen evolution. For the ligand vibration of Zn$^{2+}$-OSO$_2$-R and Zn$^{2+}$-H$_2$O in Zn(PS)$_2$ electrolyte at 240 and 390 cm$^{-1}$ respectively, the peak position of Zn$^{2+}$-OSO$_2$-R is 20 cm$^{-1}$ lower than that of Zn$^{2+}$-SO$_4^{2-}$ bond, suggesting the relative weak bonding strength. The red shift for the symmetric and asymmetric vibration of SO$_3$ demonstrates the increased number of Zn$^{2+}$-OSO$_2$-R species at higher concentrations[29]. Moreover, the peak intensity of D-rich component intensifies with the increased concentration of Zn(PS)$_2$ (Fig. 2c and Supplementary Tables 3, 4). Therefore, the Zn(PS)$_2$ electrolyte with the dominant CIP would decrease the coordinated number of H$_2$O molecules, thus inhibiting hydrogen evolution for the relatively high CE of Cu-Zn asymmetric cells (Supplementary Fig. 11). The weak interaction between PS$^-$ anion and Zn$^{2+}$ ions is helpful for the desolvation process of Zn$^{2+}$ ions, and thus lowering voltage hysteresis (Fig. 1a).

Since the dendrite growth happens at the electrode-electrolyte interface, the adsorption ability between Zn and TBA$^+$ or Zn$^{2+}$ ions is examined by ab-initio calculations. As shown in Fig. 2d, the adsorption energy of TBA$^+$ ions on zinc (002) crystal plane is −3.62 eV, higher than that of Zn$^{2+}$ ions (−2.25 eV). The preferable adsorption of TBA$^+$ ions would inhibit the 2D diffusion of Zn$^{2+}$ ions for the further dendrite growth (Fig. 2e), which is in agreement with the steady deposition current (Supplementary Fig. 10).

With the positive electrostatic potential (ESP) of PS$^-$ (−6.4 vs. −10.46 eV for SO$_4^{2-}$, Fig. 3a), the binding energies between Zn$^{2+}$-H$_2$O, Zn$^{2+}$-SO$_4^{2-}$, SO$_4^{2-}$-H$_2$O, Zn$^{2+}$-PS$^-$, and PS$^-$-H$_2$O are −104.2, −642.8, −32.1, −420.7 and −15.3 kcal mol$^{-1}$, respectively (Supplementary Fig. 21). Obviously, the electrostatic force between Zn$^{2+}$ and PS$^-$ ions is weaker than that with SO$_4^{2-}$ ions, but stronger than that between Zn$^{2+}$ ion and H$_2$O molecules. To investigate the four typical solvation structures in ZnSO$_4$ and Zn(PS)$_2$ solutions (Fig. 3b and Supplementary Fig. 22a), the corresponding radial distribution functions (RDFs) and coordination number are analyzed. In ZnSO$_4$ electrolyte, a sharp peak at 1.92 Å with the coordination number of 5.22 is corresponding to the distance between Zn$^{2+}$ ion and oxygen of water molecules. The coordination peak of Zn$^{2+}$ ions with SO$_4^{2-}$ anions appears at 1.78 Å with the coordination number of 0.77 (Supplementary Fig. 22b). Therefore, Zn$^{2+}$ ions are preferable in the form of six coordination in the solution with/without anions involved in the first solvation structure. In contrast, the coordination peaks of Zn-O$_w$ and Zn-O$_{anion}$ are located at 1.94 and 1.82 Å with the coordination number of 5.61 and 0.39, respectively in

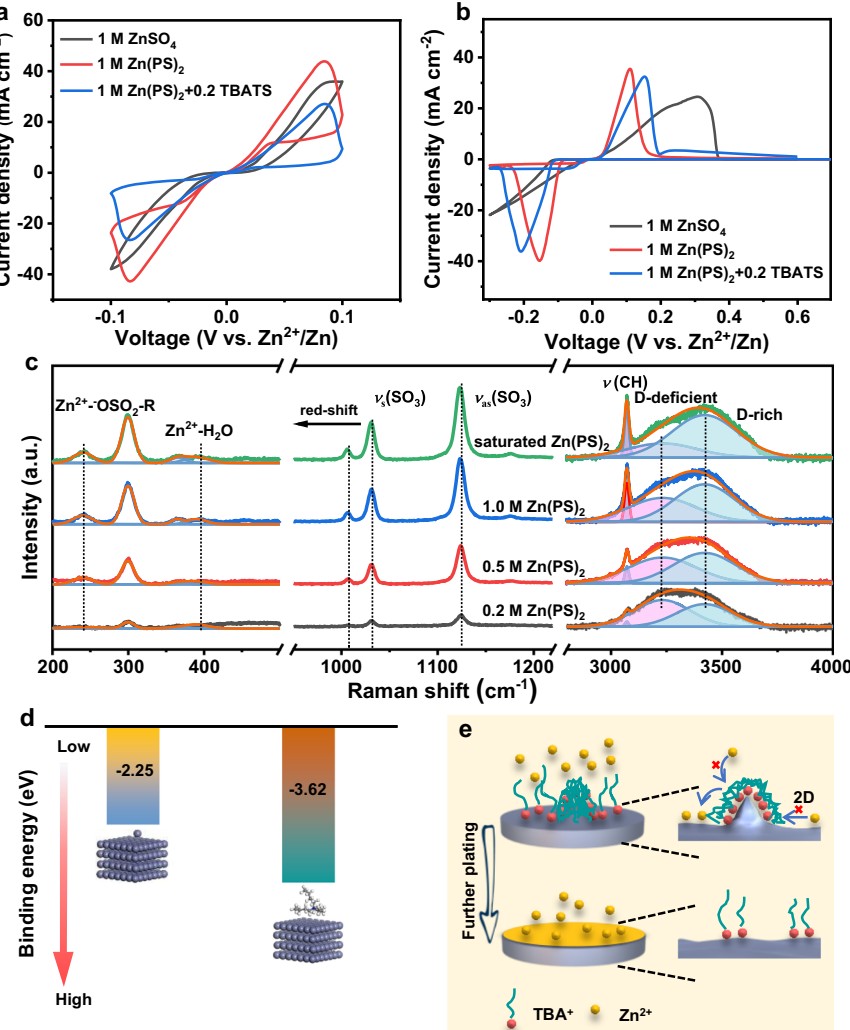

**Fig. 2 | The electrochemical and spectra analysis of electrolyte structures.** CV curves of (**a**) Zn-Zn symmetric cells and (**b**) Ti-Zn asymmetric cells in different electrolytes. **c** Raman spectra of $Zn(PS)_2$ solutions with various concentrations. **d** The adsorption energies of $Zn^{2+}$ and $TBA^+$ ions on zinc (002) crystal plane. **e** Schematic illustration the electric shielding effect of TBATS additives.

$Zn(PS)_2$ (Fig. 3c). The charge dispersed for $PS^-$ anions demonstrates the weak interaction with $Zn^{2+}$ ions, resulting in the enlarged coordination length in comparison with $SO_4^{2-}$ anions. Specifically, the desolvation energy calculated for $Zn(H_2O)_5(SO_4)$ and $Zn(H_2O)_5(PS)^+$ is 719.4 and 492.2 kcal mol$^{-1}$, respectively (Fig. 3d). Additionally, the bulky $PS^-$ anions also decrease the coordination number of $H_2O$ molecules which is in agreement with the Raman results.

The size of $PS^-$ anions is two and a half times larger than that of divalent $SO_4^{2-}$ anions (Supplementary Fig. 23). Thus, such anions would participate in the H-bonding network and decrease the $H_2O$ coordinated number. The breakage of H-bond would prevent $H_2O$ molecules from the decomposition with hydrogen evolution (Fig. 3e). To detect the disorder degree, the differential scanning calorimetry (DSC) measurement revealed that the freezing point of $Zn(PS)_2$ (−25.6 °C) is lower than that of $ZnSO_4$ solution (−19.7 °C). The hydrogen bonding number between water and anions formed in $Zn(PS)_2$ is greater than that in $ZnSO_4$ solution (Fig. 3f). The signal shift of [1]H Nuclear magnetic resonance spectroscopy ([1]H NMR) to lower magnetic fields after the addition of zinc salts suggests the enhanced de-shielding effect in 1 M $Zn(PS)_2$. The hydrogen bonds formed between anion and water molecules are stronger than that among water molecules, which weakens the shielding effect of electrons around atomic nucleus (Fig. 3g). With such robust H-bond network with $PS^-$

anions, the HER would be further inhibited via decreasing coordination with $H_2O$ molecules. (Supplementary Fig. 24).

## The electrode-electrolyte interface and function mechanism

The measured zeta potential of Zn in 1 M $ZnSO_4$ solution is −5.24 mV, significantly decreased to −1.96 mV for $Zn(PS)_2$ solution owing to the low charge of $PS^-$ anions[30]. However, the zeta potential increases to 1.37 mV after introducing TBATS. The charge polarity shift on zinc surface further demonstrates the strong electrostatic absorption of $TBA^+$ cations, which would induce rearrangement of ion distribution (Fig. 4a). The surface capacitance measurement exhibits the lower double-layer capacitance in $Zn(PS)_2$ electrolyte due to the large volume of $PS^-$ anions with increasing electric double layer (EDL) thickness. The further capacitance decrease with the addition of TBATS would be attributed to the strong adsorption of $TBA^+$ cation on the Zn surface (Fig. 4b). For the in situ Raman spectra, only electrolyte signals are detected on the Zn surface and gradually disappear with prolonging the plating time to 39 minutes (Fig. 4c) and no electrolyte decomposition product is formed, which would avoid the formation of unstable solid state interface (SEI) with the decomposition of electrolyte[31]. The fitted charge transfer resistance ($R_{ct}$) of Zn-Zn symmetric cells with $ZnSO_4$ electrolyte is 492.6 Ω at the pristine state and decreases to 292 Ω due to the activation process[32]. Lower values of

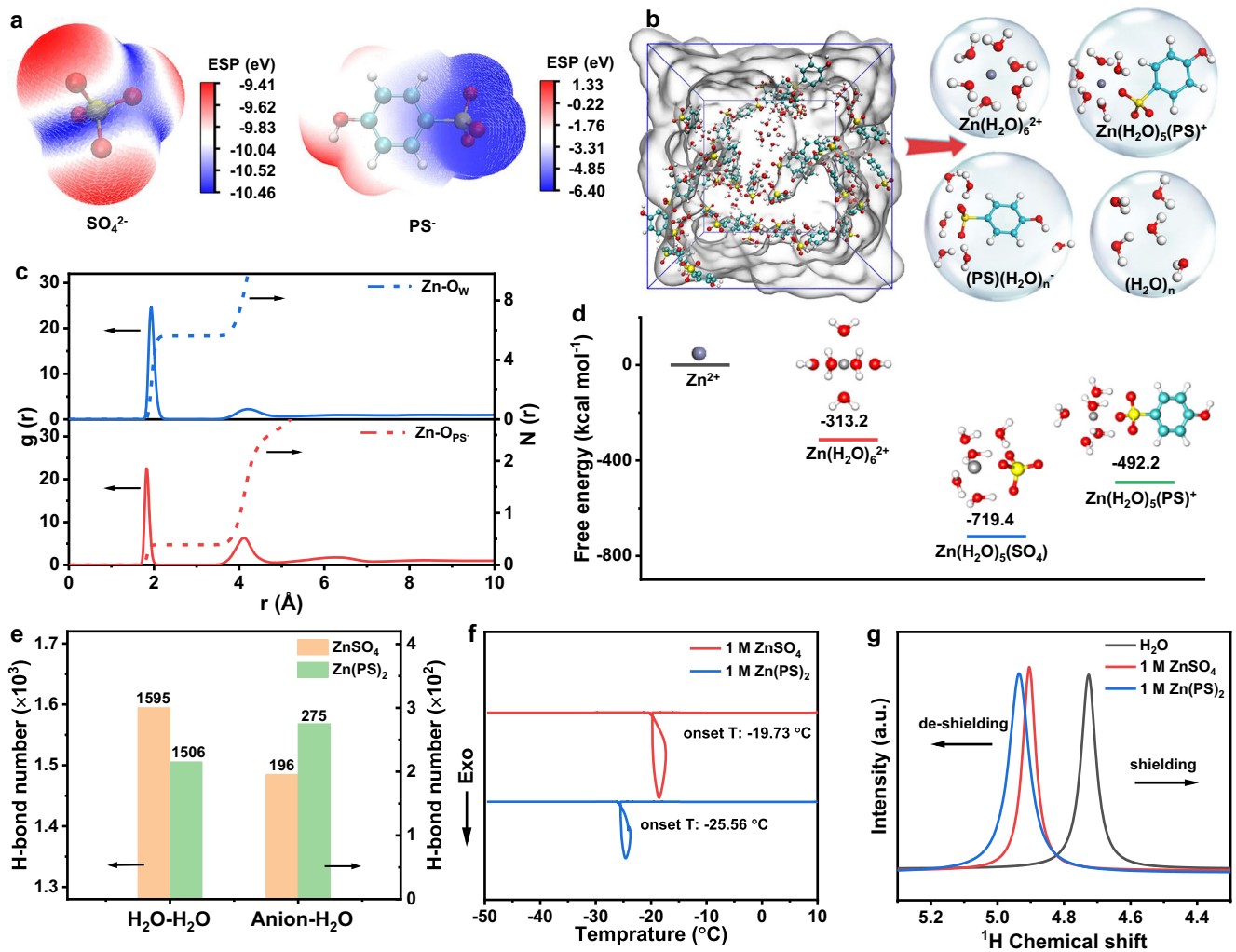

**Fig. 3 | Theoretical calculation of solvation structures.** ESP of (**a**) $SO_4^{2-}$ and PS$^-$ ions. **b** The MD snapshot of 1 M Zn(PS)$_2$ and the electrolyte structure. **c** The RDF g(r) and coordination number N(r) of Zn-O$_w$ and Zn-O$_{anion}$. **d** The desolvation energy of Zn(H$_2$O)$_5$(SO$_4$) and Zn(H$_2$O)$_5$(PS)$^+$. **e** The H-bond number in water clusters of the electrolytes in MD simulations. **f** DSC results of 1 M ZnSO$_4$ and Zn(PS)$_2$ solutions. **g** $^1$H chemical shift of H$_2$O.

261.7 and 163.5 Ω in Zn(PS)$_2$ electrolyte are ascribed to the fast Zn$^{2+}$ ion coupled electron kinetics and high ions conductivity (38.97 mS cm$^{-1}$, Supplementary Fig. 25). The R$_{ct}$ increases to 295.7 and 198.3 Ω with the addition of TBATS due to the absorption of non-electrochemically active TBA$^+$. Only one semicircle observed after 10th cycle indicates a stable electrode-electrolyte interface without the passivation layer formation (Supplementary Fig. 26). The activation energy ($E_a$) is evaluated according to the Arrhenius equation (Fig. 4d, e, Supplementary Fig. 27 and Supplementary Tables 5-7). The calculated $E_a$ for zinc plating/striping process in 1 M ZnSO$_4$, 1 M Zn(PS)$_2$ and 1 M Zn(PS)$_2$ + 0.2 TBATS electrolytes are 61.02, 50.92 and 51.97 kJ mol$^{-1}$, respectively. The reduced $E_a$ for Zn(PS)$_2$ with/without TBATS suggest the favorable desolvation process with low energy barrier. Furthermore, the Zn$^{2+}$ ion transference number ($t_{Zn^{2+}}$) was calculated to detect the ions migration behavior in different electrolyte (Supplementary Fig. 28). The $t_{Zn^{2+}}$ is only 0.25 in 1 M ZnSO$_4$ which is mainly attributed to the electrically neutral Zn(H$_2$O)$_5$(SO$_4$) with high desolvation energy. Thanks to the positively charged Zn(H$_2$O)$_5$(PS)$^+$ solvation structures, the $t_{Zn^{2+}}$ for 1 M Zn(PS)$_2$ and 1 M Zn(PS)$_2$ + 0.2 TBATS electrolytes significant increase to 0.78 and 0.76, respectively, which would weaken the concentration polarization at the electrode-electrolyte interface.

Based on the above experimental and theoretical calculation results, the plausible mechanism of the progressive electrolyte is proposed. Initially, the binding energy of solvation shell with charge dispersed PS$^-$ anions and Zn$^{2+}$ ions is low, which is favorable to the desolvation process of Zn$^{2+}$ ions for improving the transport kinetics. Secondly, bulky anions reduce the coordination number of water molecules owing to the steric hindrance effect in the solvation shell. Additionally, bulky anions are involved into the hydrogen bonding network, thus reducing the free water molecules on the Zn surface. TBA$^+$ cations could also be adsorbed on Zn surface to play a physical shielding role. Thirdly, the rapid ion transport kinetics would lower the diffusion polarization to inhibit the dendrites growth. The relative high bonding energy between TBA$^+$ ions and zinc substrate make the protrusions with accumulated electric field preferentially covered by TBA$^+$, which prevents the unfavorable deposition of zinc. Therefore, the collaborative strategy is crucial to achieve high-performance of ZIBs.

## Rechargeable batteries in such electrolyte

To demostrate the superiority of Zn(PS)$_2$ + 0.2 TBATS electrolyte in a practical battery, PANI nanofibers (~120 nm in diameter) were synthesized to fabricate the full batteries (Supplementary Fig. 29)[33]. The CV curves of such full batteries with ZnSO$_4$ electrolyte demonstrate an obvious electrochemical polarization with large peak separation of 357 mV. In contrast, the small polarizations of 196 and 206 mV are observed and another reduction peak is observed at 0.86 and 0.84 V in

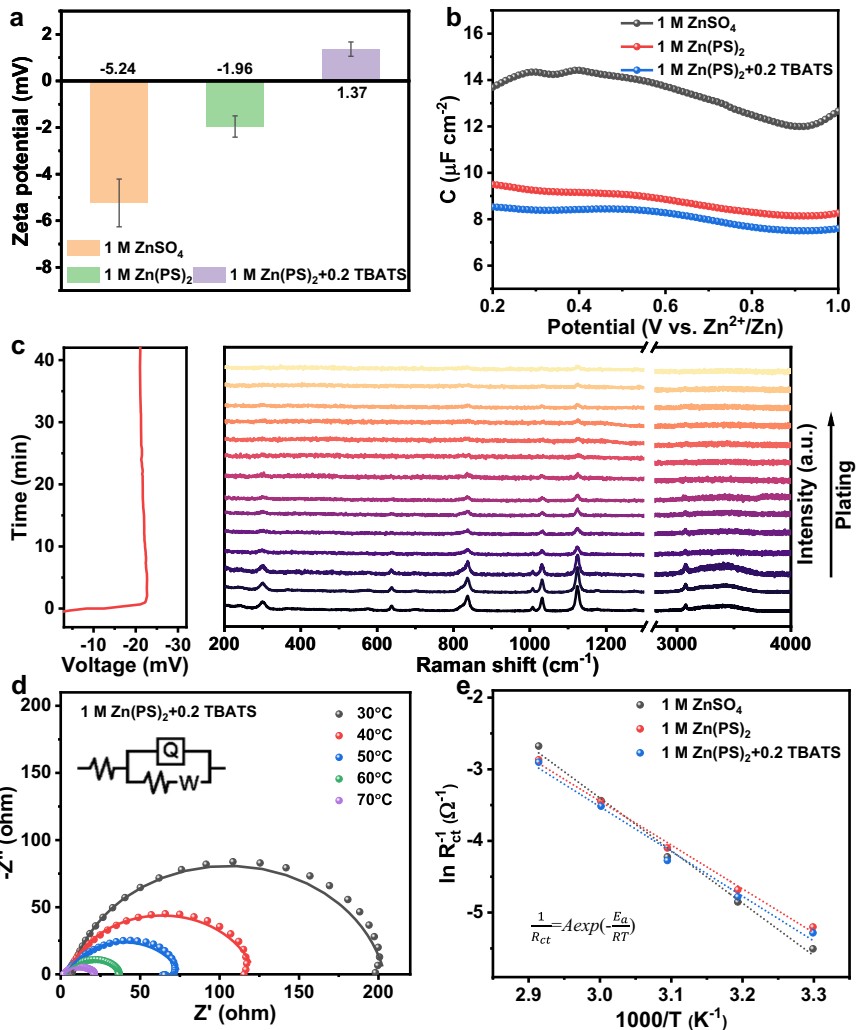

**Fig. 4 | The analysis of electrode-electrolyte interface. a** Zeta potential of zinc powder in various electrolytes. The error bars represent the standard deviation from three independent measurements. **b** Differential capacitance curves conduct in different electrolytes. **c** In situ Raman spectroscopy of zinc ions plating in 1 M $Zn(PS)_2$ + 0.2 TBATS electrolyte at 1 mA cm$^{-2}$ and corresponding voltage-time curves. **d** Nyquist plots of Zn electrode in 1 M $Zn(PS)_2$ + 0.2 TBATS electrolyte. **e** Arrhenius curves and comparison of activation energies in different electrolytes.

$Zn(PS)_2$ with/without 0.2 TBATS electrolyte, respectively, showing the good reversibility (Fig. 5a). The diffusion coefficient of $Zn^{2+}$ ion calculated are $9.53 \times 10^{-9}$, $5.22 \times 10^{-7}$ and $4.69 \times 10^{-7}$ cm$^2$ s$^{-1}$ in 1 M $ZnSO_4$, 1 M $Zn(PS)_2$ and 1 M $Zn(PS)_2$ + 0.2 TBATS, respectively (Supplementary Fig. 30). Such rapid diffusion of $Zn^{2+}$ ions in the composite electrolyte endows the favorable battery performance with small concentration polarization, in good agreement with CV results. The energy storage process is synergistically controlled by ionic diffusion and pseudocapacitance, endowing the large specific capacity and good rate performance (Supplementary Fig. 31). Specifically, the high specific capacities of 194 and 90 mAh g$^{-1}$ are achieved at the specific current of 0.1 and 5 A g$^{-1}$ with 1 M $Zn(PS)_2$ + 0.2 TBATS electrolyte, respectively (Fig. 5b, c). The capacity retention of 82% even after 10000 cycles in $Zn(PS)_2$ + 0.2 TBATS electrolyte shows the good cycling stability (Fig. 5d). The PANI-Zn batteries with 1 M $Zn(PS)_2$ + 0.2 TBATS electrolyte exhibit comparable and even better performance in comparison with the recent similar batteries (Supplementary Table 8). The charge transfer resistance of PANI-Zn batteries in $ZnSO_4$, $Zn(PS)_2$ and $Zn(PS)_2$ + 0.2 TBATS electrolyte are 191.6, 123.9 and 155.5 Ω, respectively, further showing the favorable charge transfer in $Zn(PS)_2$ (Supplementary Fig. 32). In comparison with the vertical growth of nanoplates in $ZnSO_4$ electrolyte, the smooth surface in 1 M $Zn(PS)_2$ + 0.2 TBATS electrolyte demonstrates the reversible redox

reactions of zinc (Supplementary Fig. 33). The peaks situated at 1168, 1339 and 1589 cm$^{-1}$ (Supplementary Table 9) are ascribed to C-H bending deformation of benzenoid ring, C-N$^+$ vibrations of delocalized polaronic structures and C-C stretching, respectively. Upon the charging process, the new peaks for the evolution of PANI structure at 416, 525 and 580 cm$^{-1}$ are assigned to the out-of- plane deformation of aromatic ring. The red-shift of peak at 1168 cm$^{-1}$ and the typical peaks of quinonoid structure at 1417 (C-C stretching vibrations of quinoid ring), 1492 (C = N stretching vibration), 1567 cm$^{-1}$ (C = C stretching vibrations) suggest the oxidation of PANI with the extraction process of $Zn^{2+}$ and H$^+$ ions. Upon the discharging process, the absence of the quinonoid structure with the gradual presence of the benzenoid ring suggests the good reversibility (Fig. 5e, Supplementary Fig. 34 and Supplementary Table 9). With the further detailed analysis (Supplementary Figs. 35, 36), the plausible energy storage process (Supplementary Fig. 37) suggests both H$^+$ and $Zn^{2+}$ ions from the bulk electrolyte are possibly involved into the polyaniline chains along with their redox reactions. With similar hydrogen ion concentration (~4.1 vs. ~4.4 for 1 M $ZnSO_4$), the additional zinc ions are available due to the favorable regulation of coordination structure, which are involved in the redox process of polyaniline due to the electrostatic interaction to enhance the specific capacity (Fig. 5b).

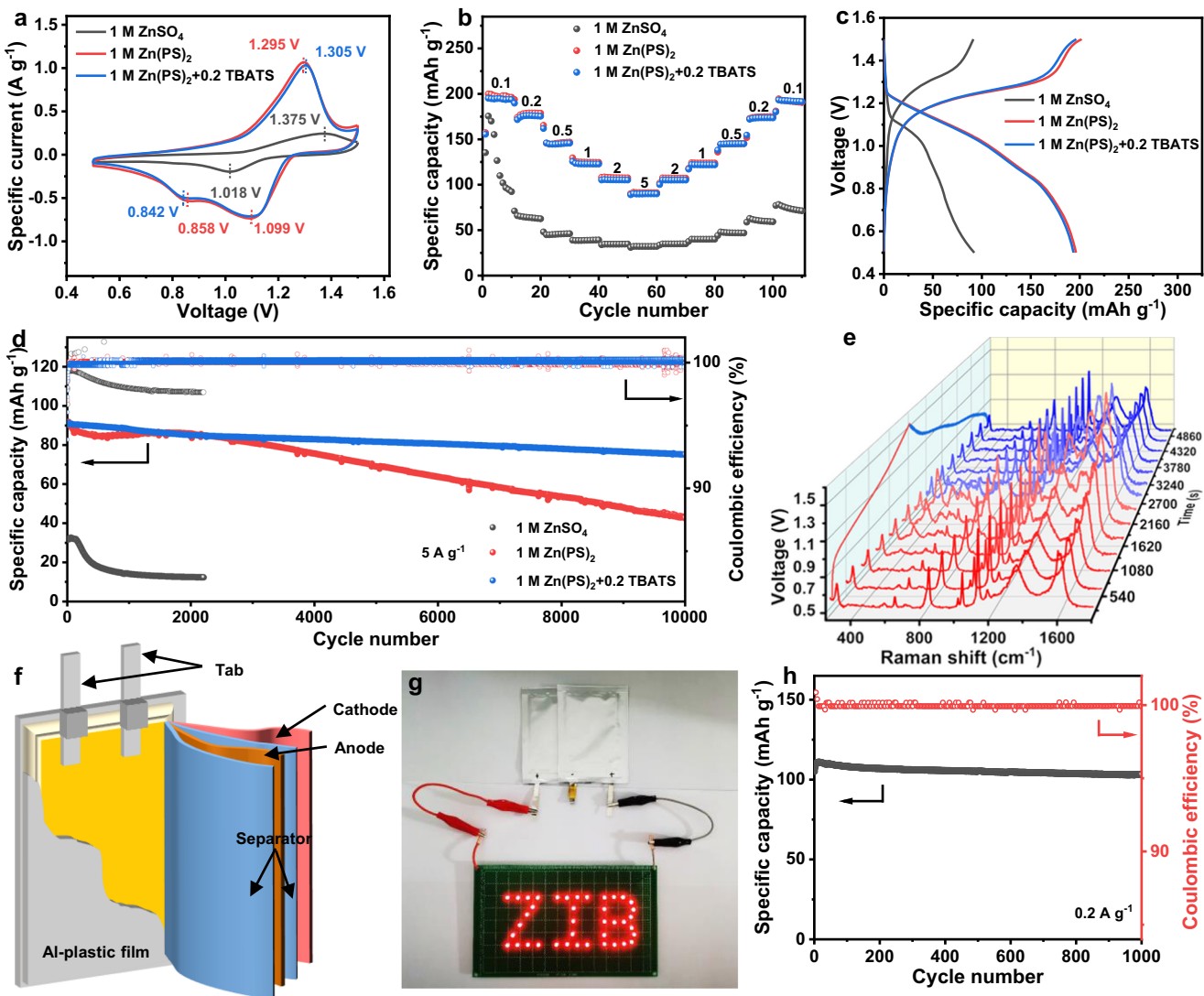

**Fig. 5 | Electrochemical performance of single cell batteries. a** Comparison of CV curves of PANI-Zn batteries at 1 mV s⁻¹ with 1 M ZnSO₄, 1 M Zn(PS)₂ and 1 M Zn(PS)₂ + 0.2 TBATS electrolytes. **b** Rate performance and **c** corresponding voltage-capacity curves at 0.1 A g⁻¹. **d** Long-term cycling stability. **e** In situ Raman spectroscopy of PANI cathode in 1 M Zn(PS)₂ + 0.2 TBATS electrolyte. **f** Schematic illustration of pouch cells. **g** Two pouch cells light 40 LEDs. **h** Cycling stability of pouch cells.

Some nanoflakes corresponding to $Zn_4SO_4(OH)_6 \cdot H_2O$ (JCPDS 39-0690, Supplementary Figs. 38 and 39) are formed on the surface of carbon cloth due to the local pH increase along with the insertion of protons[34]. Moreover, the color change of separator into yellow is attributed to the decomposition of PANI. The solvation $Zn^{2+}$ ions with charge concentrated $SO_4^{2-}$ anions would deteriorate polyaniline chain structure during the adsorption process owing to the strong coulombic force. The consumption of $H^+$ increase the concentration of $OH^-$ at interface which further combines with $SO_4^{2-}$ anions to form non-conductive $Zn_4SO_4(OH)_6 \cdot H_2O$, thus resulting in poor electrochemical performance. In contrast, no by-product is detected in $Zn(PS)_2$ electrolyte, demonstrating the steady interface environment and good structure stability of PANI in $Zn(PS)_2$ electrolyte. The practicability is also verified by fabricating pouch cells (Fig. 5f–h) to power 40 light-emitting diodes (LEDs). Moreover, the pouch cell demonstrated good cycling stability with a capacity retention of 93% for over 1000 cycles.

## Discussion

In summary, zinc phenolsulfonate with bulky and conjugated anion is proposed as a well-designed electrolyte to modulate the solvation structure of $Zn^{2+}$ ions. Such electrolyte demonstrated fast transfer kinetics and low energy barrier for the desolvation process of coordinated $Zn^{2+}$ ions. Especially, tetrabutylammonium 4-toluenesulfonate with moderate absorption ability was added as the levelling agent to inhibit the dendrite growth of zinc. Combining with DFT computation, MD modeling and spectroscopy results, the interaction of PS⁻ anions with $Zn^{2+}$ ions are revealed to adjust the coordination microenvironment for the favorable desolvation process and also suppressing hydrogen evolution via the steady hydrogen-bonding network with less water molecules cluster. The TBA⁺ cation with strong absorption ability on Zn surface exhibit the obvious shielding effect to inhibit dendrites growth and the hydrogen evolution from actived water molecules. Therefore, such electrolyte endows the good cycling stability of Zn-Zn symmetric cell over 2000 h with low overpotential of 50.2 mV, which is almost 22 times longer than that in ZnSO₄ electrolyte. Furthermore, the assembled PANI-Zn batteries exhibit the enhanced specific capacity of 194 mAh g⁻¹ and good capacity retention of 82% over 10000 cycles. This work would pave the way to enhance the reversible cycling stability of zinc-based batteries and other aqueous batteries by applying the basic principles of multifunctional electrolyte.

## Methods

### Electrolyte and electrode preparation

The $ZnSO_4$ and $Zn(PS)_2$ electrolytes were prepared by dissolving a given amount of $ZnSO_4 \cdot 7H_2O$ and $C_{12}H_{10}O_8S_2Zn$ in deionized water, respectively. The electrolytes were further regulated by adding 5, 10, 20, 50, 100 mg tetrabutylammonium 4-toluenesulfonate (TBATS) into 100 mL 1 M $Zn(PS)_2$ electrolyte and donated as 1 M $Zn(PS)_2$ + 0.05 TBATS, 1 M $Zn(PS)_2$ + 0.1 TBATS, 1 M $Zn(PS)_2$ + 0.2 TBATS, 1 M $Zn(PS)_2$ + 0.5 TBATS, 1 M $Zn(PS)_2$ + 1 TBATS, respectivily. The Zn electrode was obtained by polishing zinc foil with 5000 grit sandpaper while the Ti electrode was washed with ethanol throughtly. For the preparation of polyaniline (PANI), 0.3 mL aniline and 0.18 g ammonium peroxydisulfate (APS) were dissolved in two vitals containing 10 mL 1 M HCl, respectively. The APS solution was added into the aniline solution after cooling to 0 °C. The mixtures were left in ice water bath for 2 h. The obtained samples were washed with deionized water and then freeze dried. The cathode was prepared by coating of the slurry of 70% PANI, 20% acetylene black and 10% PVDF on carbon cloth and dried in vacuum oven for 8 h to remove residual solvent. The weight loading of PANI is about 1 ~ 1.2 mg cm$^{-2}$.

### Materials characterization

The scanning electron microscopy (SEM, Gemini SEM 300, Carl Zeiss Microscopy GmbH) was used to investigate the morphologies of samples. The X-ray diffraction method (Rigaku SmartLab 9 KW, Cu-Kα, λ = 1.5418 Å) was employed to collect the crystallographic phases of samples. Raman spectroscopy was performed on LabRAM HR Evolution (HORIBA JY) with 633 nm laser for the detection of solvation structure. Differential scanning calorimetry (DSC 250, TA Instruments) was adopted to analyze the free point of different electrolytes, and samples were scanned from 10 to −50 °C at a rate of 2 °C min$^{-1}$ under nitrogen atmosphere. 400 MHz NMR spectrometer (AVANCE400, Bruker) was applied to detect the 1H NMR spectra for different solutions with deuterated DMSO as the solvent. Zeta Sizer Malvern Nano-ZS was employed to determine the zeta potential of Zn particle in different electrolyte. The samples were prepared by the ultrasonic treating of the Zn powder in 10.0 mL electrolyte solutions to form uniformly distributed suspensions.

### Electrochemical characterization

Symmetric cells were assembled in CR 2032 coin cell using Zn foil and glass fiber as electrode and separator, respectively. The thickness of Zn foil and separator is 300 and 260 μm, respectively. The zinc foil is cut into a square electrode with a side length of 10 mm, and the diameter of circular separator is 20 mm. The three-electrode configuration was constructed with Zn foil as working electrode, the saturated Ag/AgCl as reference electrode and graphite rod as counter electrode. For Ti-Zn asymmetric cell, Ti and Zn foil were assembled for coulombic efficiency test. The cutoff voltage was set to 0.8 V. The pouch cell was assembled with PANI cathode, glass fiber separator and Zn foil anode. The PANI cathode was prepared by pressing the mixture of PANI, acetylene black and polytetrafluoroethylene (PTFE) at a weight ratio of 7:2:1 on Ti foil. The mass loading of PANI was about 70 mg and the electrolyte was 700 μL (10 μL mg$^{-1}$). The assembled batteries were examined by Neware battery test system (CT-4008-5V20mA-164) in the incubator with the fixed temperature of 28 °C. The specific capacity was calculated on the basis of the weight of PANI. For the long-term cycling test, the assembled batteries were pre-activated at the small specific current of 0.2 A g$^{-1}$ for 10 cycles. Differential capacitance-potential curves were obtained by impedance methods[35] with assembled Ti-Zn asymmetric cells. The capacitance can be calculated from the equation:

$$C = \left(2\pi f Z_{im}\right)^{-1} \qquad (1)$$

where $C$ is the capacitance, $Z_{im}$ is the imaginary component of the impedance, and $f$ is the frequency of the ac perturbation.

### Computational methods

Density Functional Theory (DFT) calculations for ESP of anions and binding energy among cations, anions and $H_2O$ were performed in Gaussian 16 W software package. Geometrical optimization and frequency analysis adopted the B3LYP[36] method with 6-311 + + G(d,p) basis sets. MD simulations for the electrolyte structures including cation / anion coordination shell, H-bond and water cluster were conducted by using the GROMACS package[37] with AMBER03 force field[38]. Water molecules were simulated with OPC model[39]. The MD parameters for $Zn^{2+}$ was based on the in-built force field parameters. The MD parameters for $SO_4^{2-}$ and $PS^-$ were generated by ACPYPE[40] and the corresponding atom charges were based on RESP charges[41]. The simulation systems contain 1000 water molecules and 18 Zn salts. Firstly, NVT run was performed at 600 K for 10 ns to accelerate the ions aggregation, and then NPT run at 298.15 K was 50 ns long for ensuring the system equilibrium. The last 10 ns was used for analysis. The calculation of H-bonds is based on the geometrical configuration with the two oxygen distance less than 3.5 Å and the O-H⋯O angle less than 30°. The RDFs were calculated by in-built module in GROMACS package. The snapshot of MD simulation is produced by VMD software.

Structure optimization was carried out using the plane-wave pseudopotential method as implemented in Castep module of the Materials Studio 2019. The generalized gradient approximation (GGA) with Perdew-Burke-Ernzerhof (PBE) exchange-correlation functional and the OTFG ultrasoft pseudopotential were used in all calculations[42,43]. The energy cutoff of plane-wave functions was 400 eV. The convergence tolerance was set to $1.0 \times 10^{-5}$ eV for the energy of per atom, 0.03 eV Å$^{-1}$ for max force, and 0.001 Å for max displacement. The binding energy ($E_b$) was calculated in four layers 7 × 7 supercell by the equation as follows:

$$E_b = E_{total} - E_{slab} - E_{molecules} \qquad (2)$$

where $E_{total}$, $E_{slab}$, and $E_{molecules}$ represent the total energy of slab contained with $Zn^{2+}$ or TBA$^+$, slab, and $Zn^{2+}$ or TBA$^+$, respectively.

### Reporting summary

Further information on research design is available in the Nature Portfolio Reporting Summary linked to this article.

## Data availability

All data that support the findings of this study are available from the corresponding author on reasonable request.

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

## Acknowledgements

J. Z. acknowledge the financial support from the National Natural Science Foundation of China (22175108), the Natural Scientific Foundation of Shandong Province (ZR2020JQ09 & ZR2022ZD27) and Taishan Scholars Program (No. tstp20221105), Project for Scientific Research Innovation Team of Young Scholar in Colleges, Universities of Shandong Province (2019KJC025). The authors also acknowledge the assistance of the Analytical Center for Structural Constituent and Physical Property of Core Facilities Sharing Platform, Shandong University and the National Synchrotron Radiation Laboratory (NSRL) at the University of Science and Technology of China.

## Author contributions

J. Z. directed the research. S. C. and J. Z. designed this work. S. C. and J. M. performed the material characterization. S. C. and D. J. carried theoretical calculation. Results were discussed with J. Z., D. J., Q. C., J. M., S. H., and the manuscript was prepared with critical inputs from all the authors.

## Competing interests

The authors declare no competing interests.
