## [Peer Review File · Nature Communications]

Coordination modulation of hydrated zinc ions to enhance redox reversibility of zinc batteriesREVIEWER COMMENTS

Reviewer #1 (Remarks to the Author):

This manuscript reports the rational regulation of hydrated zinc ions via the coordination process in the presence of zinc phenolsulfonate ($Zn(PS)_2$) and tetrabutylammonium 4-toluenesulfonate (TBATS) as a new family of aqueous electrolyte. With the in-depth analysis on the coordination structure and properties of zinc ions formed in such an electrolyte system, the deposition and dissolution were examined as the reversible metal electrode, exhibiting impressive cycling stability for 2000 h with low polarization. I believe the interesting results are publishable after solving the following questions:

1. The nucleation overpotentials of Zn in 1 M $Zn(PS)_2$ electrolytes with/without TBATS are significantly lower than that in 1 M $ZnSO_4$. How about the deposition on other substrates, such as Ti? The nucleation potential on Ti substrate in Figure 2b should be noted.
2. The deposition structure is important to exhibit the cycling stability of metal-based batteries. In-situ optical microscopy observations exhibit the cross-section image of the Zn deposition process (Figure 1i, j). However, the real morphology after deposition should be provided to examine the different macro-structures with the possible formation of Zn dendrites.
3. Does the TBATS additives change the properties of the bulk electrolyte? Raman spectra would be helpful in detecting the potential changes of the bulk electrolyte with TBATS additives. Data should be provided to investigate.
4. How to calculate the of R_{ct} and activation energy (E_a) ? Detailed information, including the fitted curves and the equations, has to be presented to understand these results for readers better.
5. The full battery with the $Zn(PS)_2$ electrolyte demonstrated the enhanced electrochemical performance than that in $ZnSO_4$ electrolyte, why? Would the diffusion rate of Zn^{2+} ions be changed to determine the final performance?

Minor issues,

1. Figure 1b, poor visibility, letters cannot be identified.
2. Figure 2d, "strong" and "weak" cannot be used to describe binding energy, "high" and "low" will be more appropriate, or "strong binding" , "weak binding".
3. Figure 5d, it is not appropriate to use a red CE axis, as the red color is actually for a cycling stability curve.

Reviewer #2 (Remarks to the Author):

The submitted work is another effort among many in recent couple of years aimed to mitigate Zn anode corrosion as well as HER issues by electrolyte engineering. This time, the authors reported an aqueous $Zn(PS)_2$ along with TBATS surface modifier as a better electrolyte than $ZnSO_4$. Note that electrolyte engineering by additives is all old tricks used by Zn-plating industry. While it makes sense to use newly designed Zn salt and additives for mitigating Zn corrosion issue (as well as HER) for Zn-plating process, it is equally important to consider its potential impact on cathodic reactions for Zn-ion batteries. Unfortunately, there is no discussion on that aspect in this paper. As many early studies have demonstrated that H^+ intercalation in cathodes accounts for a significant portion of the capacity. The source of H^+ is the H_2O solvating Zn^{2+} . If they are replaced by PS^- as stated by the authors, would H^+ storage in the cathode be inhibited and the capacity of the battery be significantly lowered? These are important questions that the authors should have addressed.

Reviewer #3 (Remarks to the Author):

In this manuscript, the authors induced the rational modulation of the coordination micro-environment with zinc phenolsulfonate ($Zn(PS)_2$) and tetrabutylammonium 4-toluenesulfonate (TBATS) as a new family of electrolytes. Such electrolytes endow the favorable coordination of zinc ions with conjugated phenolsulfonate anions that involved in hydrogen bond network to minimize

the activate water molecules, thus leading to the enhanced redox kinetics and impressive cycling stability of zinc electrode. Owing to electrochemical data and theoretical calculations, the chemical interaction of zinc ions with phenolsulfonate anions and TBATS was analyzed to provide the in-depth understanding of an improved redox reversibility. Consequently, the zinc-based batteries with redox reactions of polyaniline cathodes demonstrated the impressive cycling stability for energy storage. With the highly promising applications of zinc electrode in various zinc-based batteries, the novel electrolytes to regulate the reversible redox reactions of zinc are very intriguing candidates for next-generation energy storage. Therefore, the reviewer would recommend it to be published after suitable revisions. The comments and suggestions about this work are as follows:

1. As for the corrosion test, oxygen dissolved into the aqueous electrolyte would change the corrosion behaviors. Is oxygen removed from the aqueous electrolyte? The related information should be provided.
2. In the molecule dynamic simulation part, both $Zn(PS)_2$ and water molecules are involved in the system. Although trace amount of TBATS was added in the present electrolyte, please explain why TBATS additive is not involved in the theoretical calculation.
3. The mass loading of polyaniline in the pouch cells including the volume of electrolyte should be given in the work to understand the present electrochemical performance. Especially, the conductivity of such molecular electrolyte with coordination networks is generally not good, how to achieve the redox process with conjugated anions?
4. With the novel electrolyte of $Zn(PS)_2$ +TBATS, the full cell by coupling of zinc electrode with polyaniline demonstrated enhanced electrochemical performance. A comparable list can be added to exhibit the advances in comparison with the reported results.
5. In Figure 3d, the desolvation energy calculated for $Zn(H_2O)_6^{2+}$ is lower than those of $Zn(H_2O)_5(SO_4)$ and $Zn(H_2O)_5(PS)^+$. Accordingly, would the electrochemical performance of zinc electrodes be enhanced in the dilute solution?
6. The Zn^{2+} ion transference number calculated (Supplementary Fig. 25) for 1 M $Zn(PS)_2$ is 0.78, but slightly decreased to 0.76 with the addition of TBATS. However, the ionic conductivity of 1 M $Zn(PS)_2$ +0.2 TBATS electrolyte is better than that of the electrolyte without additives. How to correlate the relationship between the ionic conductivity and the transference number?

Point-to-Point Responses to Reviewers' Comments

We appreciate the reviewers for their helpful and insightful comments, which have greatly helped us to improve the quality of our work. The manuscript has been carefully revised accordingly and the detailed responses and quotations from the revised manuscript are listed below. All the revised parts are highlighted in blue in the manuscript.

To Reviewer#1

Reviewer #1: Comments

This manuscript reports the rational regulation of hydrated zinc ions via the coordination process in the presence of zinc phenolsulfonate ($\text{Zn}(\text{PS})_2$) and tetrabutylammonium 4-toluenesulfonate (TBATS) as a new family of aqueous electrolyte. With the in-depth analysis on the coordination structure and properties of zinc ions formed in such an electrolyte system, the deposition and dissolution were examined as the reversible metal electrode, exhibiting impressive cycling stability for 2000 h with low polarization. I believe the interesting results are publishable after solving the following questions:

We really appreciate the reviewer's positive comments.

Comments 1. The nucleation overpotentials of Zn in 1 M $\text{Zn}(\text{PS})_2$ electrolytes with/without TBATS are significantly lower than that in 1 M ZnSO_4 . How about the deposition on other substrates, such as Ti? The nucleation potential on Ti substrate in Figure 2b should be noted.

Response: Thanks. Accordingly, the nucleation potential on Ti substrate is presented in Supplementary Fig. 18, and the related discussion is added in Page 6.

“Moreover, lower nucleation potentials of 90 and 73 mV are observed on Ti substrate in 1 M $\text{Zn}(\text{PS})_2$ electrolyte with/without TBATS in comparison with the nucleation potential of 101 mV in 1 M ZnSO_4 electrolyte (Supplementary Fig. 18). The results exhibit the unique features of such electrolytes for the favorable deposition of zinc.”

Supplementary Fig. 18 Enlarged CV curves of Ti-Zn asymmetric cells in different electrolytes (Fig. 2b).

Comments 2. The deposition structure is important to exhibit the cycling stability of metal-based batteries. In-situ optical microscopy observations exhibit the cross-section image of the Zn deposition process (Figure 1i, j). However, the real morphology after deposition should be provided to examine the different macro-structures with the possible formation of Zn dendrites.

Response: Thanks for your good suggestion. The related information is added in revised manuscript (Supplementary Fig. 7), accordingly.

The obvious color difference is observed on the zinc foils in 1 M ZnSO₄ owing to the uneven deposition. With the extension of deposition time, the zinc deposited in dark gray color is formed. The gradual growth of zinc protrusion into the obvious zinc dendrites with the prolonging deposition time results in the loose deposition of Zn with porous structure and larger thickness (about 21, 16 μm) (Supplementary Fig. 8a-d). The Zn deposited in 1 M Zn(PS)₂ electrolyte is relatively uniform. In contrast, the dense deposition of zinc is observed in 1 M Zn(PS)₂+0.2 TBATS electrolyte, exhibiting the light gray color, similar with that of pure Zn. The uniform growth of Zn with the compact layer of 9.6 μm is achieved in 1 M Zn(PS)₂+0.2 TBATS electrolyte, which is close to the theoretical thickness (~ 8.5 μm under 5 mAh cm⁻², Supplementary Fig. 8e, f).

Supplementary Fig. 7 Optical images of Zn deposited at a current density of 5 mA cm⁻² in a) 1 M ZnSO₄, b) 1 M Zn(PS)₂ and c) 1 M Zn(PS)₂+0.2 TBATS electrolytes, respectively.

Supplementary Fig. 8 SEM images of a, c, e) Zn surface and the b, d, f) cross-section after zinc deposition of 5 mAh cm^{-2} in different electrolytes: a, b) 1 M ZnSO_4 , c, d) 1 M Zn(PS)_2 , e, f) $1 \text{ M Zn(PS)}_2 + 0.2 \text{ TBATS}$.

Comments 3. Does the TBATS additives change the properties of the bulk electrolyte? Raman spectra would be helpful in detecting the potential changes of the bulk electrolyte with TBATS additives. Data should be provided to investigate.

Response: Thanks. Raman spectroscopy was employed to examine the potential change of electrolytes with/without TBATS. Notably, the similar profiles of Raman spectra (Fig R1) are obtained, showing the typical features of sulfonate group and benzene ring groups. It's evident that the TBATS additive does not change basic properties of the bulk electrolyte. Considering the low amount of TBATA (0.2 mg mL^{-1}), the additive is more likely to regulate the interface of zinc/electrolyte as the shielding layer, rather than change the solvation properties of bulk electrolyte.

Fig. R1. Raman spectroscopy of 1 M Zn(PS)₂ and 1 M Zn(PS)₂+0.2 TBATS.

Comments 4. How to calculate the of R_{ct} and activation energy (E_a)? Detailed information, including the fitted curves and the equations, has to be presented to understand these results for readers better.

Response: Thank you for your kind reminder. The related information is added in revised manuscript (Fig. 4d, e, Supplementary Fig. 27 and Tables 3-5).

The EIS (Supplementary Fig. 27) were performed in three-electrode system with Zn foil as work electrode, Ag/AgCl as reference electrode, Pt plate as counter electrode in 1 M ZnSO₄, 1 M Zn(PS)₂ electrolytes with/ without TBATS, respectively. In the temperature range of 30 to 70°C with a step of 10°C, the plots of $\ln(R_{ct}^{-1})$ vs. $1/T$ (Figure 4e) were obtained from the EIS results according to the Arrhenius equation:

$$1/R_{ct} = A \exp(E_a/RT)$$

R_{ct} is the charge transfer resistance, E_a is the activation energy, T is the absolute temperature, R is the gas constant, and A is the pre-exponential factor.

The fitted results are listed in Supplementary Tables 3-5

Supplementary Fig. 27 Electrochemical impedance spectroscopy (EIS) results with changing temperatures in a) 1 M ZnSO₄ and b) 1 M Zn(PS)₂ electrolytes.

Fig. 4 (d) Nyquist plots of Zn electrode in 1 M Zn(PS)₂+0.2 TBATS electrolyte. (e) Arrhenius curves and comparison of activation energies in different electrolytes.

Supplementary Table 3 The fitted R_{ct} of Zn symmetric cells in 1 M ZnSO₄ electrolyte.

T (°C)	R_{ct} (Ω)	Error (%)	$\ln (R_{ct}^{-1})$ (Ω ⁻¹)
30	246.10	4.42	-5.50
40	128.40	3.64	-4.85
50	68.25	2.01	-4.22
60	31.16	0.30	-3.44
70	14.53	1.37	-2.68

Supplementary Table 4 The fitted R_{ct} of Zn symmetric cells in 1 M Zn(PS)₂ electrolyte.

T (°C)	R_{ct} (Ω)	Error (%)	$\ln (R_{ct}^{-1})$ (Ω ⁻¹)
30	181.50	3.14	-5.20
40	108.01	3.25	-4.68
50	60.25	0.49	-4.10
60	31.53	1.40	-3.45
70	17.61	1.32	-2.87

Supplementary Table 5 The fitted R_{ct} of Zn symmetric cells in 1 M Zn(PS)₂+0.2 TBATS electrolyte.

T (°C)	R_{ct} (Ω)	Error (%)	$\ln (R_{ct}^{-1})$ (Ω ⁻¹)
30	196.92	2.12	-5.28
40	119.03	2.62	-4.78
50	71.74	2.19	-4.27
60	33.63	0.70	-3.52
70	18.18	1.02	-2.90

Comments 5. The full battery with the Zn(PS)₂ electrolyte demonstrated the enhanced electrochemical performance than that in ZnSO₄ electrolyte, why? Would the diffusion rate of Zn²⁺ ions be changed to determine the final performance?

Response: Thanks. The diffusion coefficient of Zn²⁺ ions in different electrolytes were calculated according to the Randle-Sevcik equation:

$$I_p = 2.69 \times 10^5 n^{3/2} A D_{Zn}^{1/2} v^{1/2} C_{Zn}$$

where I_p is the peak current, n is the number of electrons, A is the area of the electrode, D_{Zn} is the diffusion coefficient of Zn²⁺ ions, v is the scan rate and C_{Zn} is the concentration of Zn²⁺ ions. Taking the average value of anodic and cathodic process, the D_{Zn} values are calculated to be 9.53×10^{-9} , 5.22×10^{-7} and 4.69×10^{-7} cm² s⁻¹ in 1 M ZnSO₄, 1 M Zn(PS)₂ and 1 M Zn(PS)₂+0.2 TBATS, respectively. The enhanced transfer ability of Zn²⁺ ions would be responsible for the good rate performance and high specific capacity. Additionally, it has been revealed that the intercalation of counterions including H⁺ is involved in energy storage process in the cathode along with the redox reactions of polyaniline (*Angew. Chem. Int. Ed.* **57**, 16359-16363 (2018); *Chem. Eng. J.* **448**, 137711 (2022)). In the present case, both H⁺ and Zn²⁺ as counterions would be involved in the energy storage process of polyaniline due to the electrostatic interaction. Notably, the additional zinc ions with the enhanced transfer ability are available in Zn(PS)₂+0.2 TBATS electrolyte due to the easy desolvation, which would improve the energy storage process of polyaniline with counterion intercalation and thus enhance electrochemical performance.

Supplementary Fig. 30 Linear fitting between the peak current and the square root of the scan rates of the CV curves for PANI-Zn batteries.

Minor issues,

Comments 1. Figure 1b, poor visibility, letters cannot be identified.

Response: Thank you for your kind reminder. The Fig. 1b is revised to improve the visibility.

Fig. 1b Comparison of cycling stability with recently reported results at different current density and capacity.

Comments 2. Figure 2d, “strong” and “weak” cannot be used to describe binding energy, “high” and “low” will be more appropriate, or “strong binding”, “weak binding”.

Response: Thank you for your good suggestion. The Fig. 2d is revised, accordingly.

Fig. 2d The adsorption energies of Zn²⁺ and TBA⁺ ions on zinc (002) crystal plane.

Comments 3. Figure 5d, it is not appropriate to use a red CE axis, as the red color is actually for a cycling stability curve.

Response: Thanks. The Fig. 5d is corrected.

Fig. 5d Long-term cycling stability.

To Reviewer#2

Reviewer #2: The submitted work is another effort among many in recent couple of years aimed to mitigate Zn anode corrosion as well as HER issues by electrolyte engineering. This time, the authors reported an aqueous Zn(PS)₂ along with TBATS surface modifier as a better electrolyte than ZnSO₄. Note that electrolyte engineering by additives is all old tricks used by Zn-plating industry. While it makes sense to use newly designed Zn salt and additives for mitigating Zn corrosion issue (as well as HER) for Zn-plating process, it is equally important to consider its potential impact on cathodic reactions for Zn-ion batteries. Unfortunately, there is no discussion on that aspect in this paper. As many early studies have demonstrated that H⁺ intercalation in cathodes accounts for a significant portion of the capacity. The source of H⁺ is the H₂O solvating Zn²⁺. If they are replaced by PS⁻ as stated by the authors, would H⁺ storage in the cathode be inhibited and the capacity of the battery be significantly lowered? These are important questions that the authors should have addressed.

Response: Thanks for the reviewer's positive comments and good question.

We agree with the reviewer that the electrolyte additives have been used in the metal plating industry and to inhibit the metal corrosion. Indeed, the related research works would provide useful principles for rationally designing new electrolytes to regulate the deposition of metals. The development of electrolyte as an important component of a battery is also crucial to achieve the high-performance (*Nat. Energy* **6**, 763-763 (2021)).

We really appreciate the reviewer's deep insight with the specific focus on the energy storage process of polyaniline in the cathode. We agree with the reviewer that counterions (e.g., H⁺) are involved in the energy storage process of polyaniline due to the electrostatic interaction along with the redox reactions (*Angew. Chem. Int. Ed.* **57**, 16359-16363 (2018); *Chem. Eng. J.* **448**, 137711 (2022)). The related discussion is added in Page 14 and Supplementary information (Pages 37-38).

According to the plausible energy storage process (Supplementary Fig. 36), both H⁺ ion and Zn²⁺ ions from the bulk electrolyte are possibly involved into the polyaniline chains along with the redox reactions during the discharge process. During the charging process, the counterions, such as SO₄²⁻, PS⁻ anions are also adsorbed into the polyaniline chains to balance the charge. The adsorption of counterions on the random polyaniline chains is highly dependent on the electrostatic interaction. To examine the potential impact on cathodic reactions for Zn-ion batteries, the similar profiles of Raman spectra (Fig. R1) in 1 M Zn(PS)₂+0.2 TBATS and 1 M ZnSO₄ show the gradual changes of characteristic groups corresponding to the benzene ring and the quinone ring during the charge and discharge process. Upon the charging process, the new peaks for the evolution of PANI structure at 416, 525 and 580 cm⁻¹ are assigned to the out-of- plane

deformation of aromatic ring. The red-shift of peak at 1168 cm^{-1} (C-H bending vibrations of quinone ring) and the typical peaks of quinonoid structure at 1417 (C-C stretching vibrations of quinoid ring), 1492 (C=N stretching vibration), 1567 cm^{-1} (C=C stretching vibrations) suggest the oxidation of PANI with the extraction process of Zn^{2+} and H^+ ions. Upon the discharging process, the absence of the quinonoid structure with the gradual presence of the benzenoid ring suggests the good reversibility (Fig. 5e, Supplementary Fig. 34 a,b and Table 7). The results revealed that the energy storage mechanism is not changed obviously in the different electrolytes.

The pH of ~ 4.1 for 1 M Zn(PS)_2 , $1\text{ M Zn(PS)}_2+0.2\text{ TBATS}$ is comparable with that for 1 M ZnSO_4 (~ 4.4). For the different electrolytes with similar hydrogen ion concentration, the possible H^+ intercalation is not changed during the energy storage process of polyaniline in cathode, although the introduction of large anion decreases the coordination number of water molecules that would dissociate into the hydrogen ion at the close interface of zinc. Especially, the enhanced battery performance with larger specific capacity (Fig. 5b) suggests that the presence of PS^- does not result in the lack of H^+ for capacity fade. In the presence case, both H^+ and Zn^{2+} as counterions would be involved in the energy storage process along with the redox of polyaniline chains. Notably, the low binding energy of the Zn-PS bond ($-420.7\text{ kcal mol}^{-1}$) in $1\text{ M Zn(PS)}_2+0.2\text{ TBATS}$ suggests the easy desolvation into the free Zn^{2+} ions for the energy storage process. The diffusion coefficient of Zn^{2+} ions (D_{Zn}) are calculated to be 9.53×10^{-9} and $4.69\times 10^{-7}\text{ cm}^2\text{ s}^{-1}$ in 1 M ZnSO_4 and $1\text{ M Zn(PS)}_2+0.2\text{ TBATS}$, respectively (Supplementary Fig. 30). The enhanced Zn^{2+} ions migration ability would also contribute to the energy storage process of polyaniline with cation intercalation.

From the calculation results of Zn^{2+} ion transference number ($t_{\text{Zn}^{2+}}$, Supplementary Fig. 28), the $t_{\text{Zn}^{2+}}$ for $1\text{ M Zn(PS)}_2+0.2\text{ TBATS}$ is 0.76 , which is much higher than 0.25 in 1 M ZnSO_4 , indicating that more Zn^{2+} ions can be migrated to supplement the consumption at the electrode-electrolyte interface.

The XPS spectra were performed to examine the component changes during the PANI redox process. As shown in Supplementary Fig. 35a, C, N and Cl elements were detected in the initial PANI, suggesting the successful preparation of PANI. With the absence of Cl^- dopants after the first discharge cycle (Supplementary Fig. 35b), the peaks (Supplementary Fig. 35c, d), corresponding to Zn and S elements respectively are observed, suggesting Zn^{2+} , PS^- ions are involved in the energy storage process. $\text{H}^+/\text{Zn}^{2+}$ as counterions, are adsorbed onto the reduced polyaniline chains to neutralize the charges on the polyaniline chains. Typically, the oxidized groups on the polyaniline chains, such as $-\text{NH}^+$ and $-\text{NH}^{2+}$, are stabilized by the PS^- anion via the electrostatic interaction. The N 1s can be fitted with four components, i.e., $-\text{N}=\text{}$ ($\sim 398.6\text{ eV}$), $-\text{NH}-$ ($\sim 399.5\text{ eV}$), $-\text{NH}^+$ ($\sim 400.6\text{ eV}$) and $-\text{NH}^{2+}$ ($\sim 402\text{ eV}$). The $-\text{NH}-$ component is

corresponding to the reduced state and the others are in the oxidized status. The amount of the oxidized and reduced components are almost equal in accord with the emeraldine state of the polymer at the initial state (Supplementary Fig. 35e, f). Upon the discharge process, the reduced -NH- component increases to 70.5% with the H^+/Zn^{2+} adsorption. Upon the charge process, polyaniline is oxidized, H^+/Zn^{2+} is desorbed from the polyaniline chains with the decreasing of the reduced -NH- component (22.7%). Meanwhile, the oxidized components, -NH⁻ and -NH⁼ increase to 49.5 and 19.1%, respectively, accompanied by the adsorption of dopant PS^- to balance charge.

On the basis of the above discussion, the H^+ intercalation in cathode would not be changed in the present electrolytes with similar pH values. The intercalation of both ions including H^+ and Zn^{2+} is coupled with the redox reactions of polyaniline. The calculated desolvation energy of $Zn(H_2O)_5(PS)^+$ and $Zn(H_2O)_5(SO_4)$ are $-492.2 \text{ kcal mol}^{-1}$ and $-719.4 \text{ kcal mol}^{-1}$, respectively (Fig. 3d), which is favorable to the desolvation process of Zn^{2+} ions. Notably, the additional zinc ions are available, which would be involved in the energy storage process of polyaniline with counterion intercalation, thus leading to the improved specific capacity (Supplementary Fig. 36).

For the anode, the theoretical understanding and in-situ spectroscopy analysis revealed that the favorable coordination of hydrated zinc ions with the conjugated PS^- anions are able to minimize the activate water molecules, thus improving the zinc/electrolyte interface stability (Fig. R2). Therefore, the present electrolyte does not result in the lack of H^+ for capacity fade, but enhance the battery performance with the favorable regulation on the coordination of hydrated zinc ions.

Fig. R1 In-situ Raman spectra of PANI cathodes in **a** 1 M $ZnSO_4$ and **b** 1 M $Zn(PS)_2+0.2$ TBATS electrolyte.

Supplementary Fig. 30 Linear fitting between the peak current and the square root of the scan rates of the CV curves for PANI-Zn batteries.

Supplementary Fig. 35 a) The full XPS spectra and core-level spectra of b) Cl 2p, c) Zn 2p, d) S 2p, e) N 1s of PANI at various status in 1 M Zn(PS)₂+0.2 TBATS electrolyte. f) the corresponding changes of N components.

Supplementary Fig. 36 The proposed energy storage process of PANI in a) Zn(PS)₂ and b) ZnSO₄ electrolytes.

Fig. R2 Schematic illustrations for the possible electric double layer (EDL) structure of the close interface of Zn anode in a) ZnSO₄ and b) Zn(PS)₂ electrolytes.

To Reviewer#3

Reviewer #3: In this manuscript, the authors induced the rational modulation of the coordination micro-environment with zinc phenolsulfonate ($Zn(PS)_2$) and tetrabutylammonium 4-toluenesulfonate (TBATS) as a new family of electrolytes. Such electrolytes endow the favorable coordination of zinc ions with conjugated phenolsulfonate anions that involved in hydrogen bond network to minimize the activate water molecules, thus leading to the enhanced redox kinetics and impressive cycling stability of zinc electrode. Owing to electrochemical data and theoretical calculations, the chemical interaction of zinc ions with phenolsulfonate anions and TBATS was analyzed to provide the in-depth understanding of an improved redox reversibility. Consequently, the zinc-based batteries with redox reactions of polyaniline cathodes demonstrated the impressive cycling stability for energy storage. With the highly promising applications of zinc electrode in various zinc-based batteries, the novel electrolytes to regulate the reversible redox reactions of zinc are very intriguing candidates for next-generation energy storage. Therefore, the reviewer would recommend it to be published after suitable revisions. The comments and suggestions about this work are as follows:

We really appreciate the reviewer's positive comments.

Comments 1. As for the corrosion test, oxygen dissolved into the aqueous electrolyte would change the corrosion behaviors. Is oxygen removed from the aqueous electrolyte? The related information should be provided.

Response: Thanks for your good question. In order to eliminate the influence of dissolved oxygen in the electrolyte, nitrogen is bubbled continuously into the electrolyte for 30 min before the corrosion test. Then, zinc foil was put into the electrolyte and sealed in the glass vial for one week.

Supplementary Fig. 12 Optical image of glass vial for the corrosion test.

Comments 2. In the molecule dynamic simulation part, both $Zn(PS)_2$ and water molecules are involved in the system. Although trace amount of TBATS was added in

the present electrolyte, please explain why TBATS additive is not involved in the theoretical calculation.

Response: Thanks for your good question. The solvation structure of Zn^{2+} ions was simulated theoretically to obtain the bond information with large cations in electrolyte on the basis of the present computing ability available. Such information is helpful to understand the coordination of Zn^{2+} ions with water molecules. In this work, the optimum electrolyte composition is 1 M $Zn(PS)_2$ with 0.2 mg mL^{-1} TBATS. The calculated molar ratio of zinc salt over additive is about 2068: 1. Therefore, the theoretical calculation with the trace additives involved was established in order to optimize the calculation model and calculation resources.

Comments 3. The mass loading of polyaniline in the pouch cells including the volume of electrolyte should be given in the work to understand the present electrochemical performance. Especially, the conductivity of such molecular electrolyte with coordination networks is generally not good, how to achieve the redox process with conjugated anions?

Response: Thanks. The related information is added in the revised manuscript (Page 17).

“The pouch cell was assembled with PANI cathode, glass fiber separator and Zn foil anode. The PANI cathode was prepared by pressing the mixture of PANI, acetylene black and polytetrafluoroethylene (PTFE) at a weight ratio of 7:2:1 on Ti foil. The mass loading of PANI was about 70 mg and the electrolyte was $700 \text{ }\mu\text{L}$ ($10 \text{ }\mu\text{L mg}^{-1}$).”

The ion conductivity of 1 M $Zn(PS)_2$ calculated is 38.97 mS cm^{-1} which is close to the reported value of 1 M $ZnSO_4$ (43.7 mS cm^{-1} , *Angew. Chem. Int. Ed.* **60**, 18247-18255 (2021)). When 0.2 mg mL^{-1} TBATS is added, the conductivity further increases to 41.94 mS cm^{-1} . The results exhibit that the ion conductivity of the new electrolyte is even better than that of 1 M $ZnSO_4$. Especially, the calculated desolvation energy of $Zn(H_2O)_5(PS)^+$ is $-492.2 \text{ kcal mol}^{-1}$ (Fig. 3d), which is lower than that of $Zn(H_2O)_5(SO_4)$ ($-719.4 \text{ kcal mol}^{-1}$). The easily available of zinc ions with good transfer ability would contribute to the charge storage process, as demonstrated in Supplementary Fig. 36.

Comments 4. With the novel electrolyte of $Zn(PS)_2$ +TBATS, the full cell by coupling of zinc electrode with polyaniline demonstrated enhanced electrochemical performance. A comparable list can be added to exhibit the advances in comparison with the reported results.

Response: Thanks. The comparison table is added in the revised manuscript (Supplementary Table 6). It can be seen that the fabricate PANI-Zn batteries with 1 M $Zn(PS)_2$ +0.2 TBATS electrolyte exhibit the better performance in comparison with the

recent reports with other electrolytes.

Supplementary Table 6. The electrochemical performance comparison of PANI-Zn batteries based on different electrolytes.

Electrolyte	Specific capacity	Capacity retention	Ref.
1 M Zn(PS) ₂ +0.2 TBATS	194 mAh g ⁻¹ at 0.1 A g ⁻¹	82% after 10000 cycles	This work
9 M ZnCl ₂	183 mAh g ⁻¹ at 0.7 A g ⁻¹	83.5% after 1000 cycles	[1]
1 M ZnSO ₄ +4 M EMImCl	154.4 mAh g ⁻¹ at 1 A g ⁻¹	78.8% after 300 cycles	[2]
ZnCl ₂ : EG = 1: 4 (molar ratio)	180 mAh g ⁻¹ at 0.1 A g ⁻¹	78% after 10000 cycles	[3]
1 M Zn(CF ₃ SO ₃) ₂	191 mAh g ⁻¹ at 0.05 A g ⁻¹	92% after 3000 cycles	[4]
2 M ZnSO ₄ +0.05 M SG	192.3 mAh g ⁻¹ at 1 A g ⁻¹	81.7% after 1400 cycles	[5]
7.5 M ZnCl ₂	106.2 mAh g ⁻¹ at 0.02 A g ⁻¹ (-70°C)	~100% after 2000 cycles (-70°C)	[6]

[1] *Chem. Commun.* **58**, 1693-1696 (2022); [2] *Angew. Chem. Int. Ed.* **60**, 23357-23364 (2021); [3] *Angew. Chem. Int. Ed.* **61**, e202206717 (2022); [4] *Adv. Funct. Mater.* **28**, 1804975 (2018); [5] *Adv Mater* **34**, e2206963 (2022); [6] *Nat. Commun.* **11**, 4463 (2020).

Comments 5. In Figure 3d, the desolvation energy calculated for Zn(H₂O)₆²⁺ is lower than those of Zn(H₂O)₅(SO₄) and Zn(H₂O)₅(PS)⁺. Accordingly, would the electrochemical performance of zinc electrodes be enhanced in the dilute solution?

Response: Thanks for your good question. According to the desolvation energy, Zn(H₂O)₆²⁺ would be more favorable for the ion transfer in the dilute solution. However, more active water molecules would be released in the Zn²⁺ ion desolvation process, resulting in the corrosion of zinc electrode along with the hydrogen evolution reaction. Therefore, additional electrolyte is added to regulate the solvation structure for improving the cycling stability. In the present case, the dilute solution 0.2 M Zn(PS)₂ exhibits the lower ions conductivity of 17.15 mS cm⁻¹ (vs. 38.97 mS cm⁻¹ for 1 M Zn(PS)₂), which is not able to meet the ion consumption in the redox reaction process, resulting in the concentration polarization in the dilute solution (Supplementary Fig. 2). Therefore, the electrochemical performance of zinc electrodes is not enhanced in the dilute solution.

Supplementary Fig. 2 a) Voltage-time curves of Zn-Zn symmetric cells in Zn(PS)₂ electrolytes with various concentrations at 1 mA cm⁻², 1 mAh cm⁻² and b) the enlarged detail.

Comments 6. The Zn²⁺ ion transference number calculated (Supplementary Fig. 25) for 1 M Zn(PS)₂ is 0.78, but slightly decreased to 0.76 with the addition of TBATS. However, the ionic conductivity of 1 M Zn(PS)₂+0.2 TBATS electrolyte is better than that of the electrolyte without additives. How to correlate the relationship between the ionic conductivity and the transference number?

Response: Thanks for your good question.

The ion transference number is the ratio of the charge transferred by the given ion to the total charge (*J. Electrochem. Soc.* **162**, A2720-A2722 (2015)). With the addition of TBATS, the ionized ions are increased in the solution due to the presence of TBA⁺ and TS⁻, which are involved in the charge transfer process. Hence, the ion transference number of Zn²⁺ ion slightly decreases to 0.76.

Ionic conductivity refers to the conduction phenomenon caused by ion migration in the electric field. The ionic conductivity of strong electrolyte solution increases with increasing concentration (the number of conductive particles) (*J. Phys. Chem. B* **105**, 4603-4610 (2001)). The number of ions that could migrate in the electric field increases because of the ionization of TBATS. Hence, the ionic conductivity of 1 M Zn(PS)₂ electrolyte increases to 41.9 mS cm⁻¹ from 39.0 mS cm⁻¹ in the presence of TBATS.

For the Zn(PS)₂ electrolyte, the Zn²⁺ ion transference number is related to the diffusion of Zn²⁺ ion and its counterion according to the following equation:

$$t_{\text{Zn}^{2+}} = \frac{2\sigma_{\text{Zn}^{2+}}}{2\sigma_{\text{Zn}^{2+}} + \sigma_{\text{PS}^{-}}}$$

where $t_{\text{Zn}^{2+}}$ is the Zn²⁺ transference number, $\sigma_{\text{Zn}^{2+}}$ is the Zn²⁺ ions conductivity, and $\sigma_{\text{PS}^{-}}$ is the PS⁻ ions conductivity. The transference number can be considered as simply the fraction of the total ionic conductivity that is carried by Zn²⁺.

REVIEWER COMMENTS

Reviewer #1 (Remarks to the Author):

The paper has been well revised and improved. Now it can be accepted as it is.

Reviewer #2 (Remarks to the Author):

The responses from the authors to this reviewer's early comments are not satisfactory. Therefore, it prompted this reviewer to read the paper for the second time. The authors' responses provided new XPS data showing the oxidation state of H⁺ of PANI during discharge (reduction) and charge (oxidation), suggesting PANI is redox reversible. But it doesn't support that there is co-insertion of H⁺ from the electrolyte, which could be a reason why PANI is a low capacity cathode. The authors could cite literature data or perform additional tests with polar aprotic solvents to prove the lack of H⁺ insertion in PANI.

The second read through also finds more issues.

- The experimental section lacks details. For example, the authors presented the data of surface zeta-potential and differential capacitance in Fig. 4, but there is virtually no description on how these data were collected.
- The overpotential of 22.6 mV of Zn/Zn symmetrical cell at 1 mA/cm² should not be considered low as the authors claimed. The author can easily compare literature data with their own data.
- The dissolution issue of PANI in aqueous solutions needs to be discussed.
- It is unclear why the authors selected PANI as the cathode since this is a low-capacity cathode. Have the authors tried high-capacity layered oxides such as V-oxides based materials?
- Fig. S5, y-axis, should the unit be mV?
- There are many other typos that should be double checked and corrected.

Reviewer #3 (Remarks to the Author):

The authors have addressed all comments very well, the paper is ready to be accepted.

Point-to-Point Responses to Reviewers' Comments

We appreciate the reviewers for their helpful and insightful comments, which have greatly helped us to improve the quality of our work. The manuscript has been carefully revised accordingly and the detailed responses and quotations from the revised manuscript are listed below. All the revised parts are highlighted in blue in the manuscript.

To Reviewer#1

Reviewer #1 (Remarks to the Author):

The paper has been well revised and improved. Now it can be accepted as it is.

We really appreciate the reviewer's support and recommendation of our work for publication in Nature Communications.

To Reviewer#2

Reviewer #2: (Remarks to the Author):

The responses from the authors to this reviewer's early comments are not satisfactory. Therefore, it prompted this reviewer to read the paper for the second time. The authors' responses provided new XPS data showing the oxidation state of H⁺ of PANI during discharge (reduction) and charge (oxidation), suggesting PANI is redox reversible. But it doesn't support that there is co-insertion of H⁺ from the electrolyte, which could be a reason why PANI is a low capacity cathode. The authors could cite literature data or perform additional tests with polar aprotic solvents to prove the lack of H⁺ insertion in PANI.

Response: We would like to thank the reviewer for his/her reading and insightful feedback. Following the valuable comments, we have revised our manuscript carefully and made substantial revisions as per the comments.

Following your valuable comments, the Zn(PS)₂ salt and TBATS additive were dissolved in an aprotic solvent, N, N-dimethylformamide (DMF) to eliminate the H⁺ insertion contribution. In the aprotic electrolyte, the CV curves exhibit a pair of redox peaks situated at 1.29 and 0.95 V, corresponding to the redox reactions of PANI (Supplementary Fig. 36a). The low ionic conductivity results in the large potential polarization (Supplementary Fig. 36b, c). In contrast, the smaller polarization in aqueous electrolyte would be contributed to the favorable redox process of PANI. The assembled PANI-Zn battery delivers the similar specific capacity at different current density (Supplementary Fig. 36d). Specifically, the specific capacity is 185 mAh g⁻¹ at the current density of 0.1 A g⁻¹ which is comparable to that in aqueous electrolyte (194 mAh g⁻¹) (Supplementary Fig. 36d).

We thanks the reviewer for proposing the valuable method to prove the possible H⁺ insertion by using an aprotic solvent. Obviously, the H⁺ insertion is not dominant to the energy storage process of PANI in the present case. It has been revealed that the counterions including H⁺ but not limited to, are involved in the energy storage process

of polyaniline due to the electrostatic interaction along with the redox reactions [e.g., *Angew. Chem. Int. Ed.* **57**, 16359-16363 (2018)]. Hence, Zn^{2+} ions as counterions (e.g., H^+) are contributed more to the specific capacity of PANI in the manuscript. The lower desolvation energy calculated is $-492.2 \text{ kcal mol}^{-1}$ for $Zn(H_2O)_5(PS)^+$ (vs. $-719.4 \text{ kcal mol}^{-1}$ for $Zn(H_2O)_5(SO_4)$) (Fig. 3d), which would contribute to the favorable desolvation process of Zn^{2+} ions. Therefore, the additional zinc ions are involved in the energy storage process of polyaniline with counterion intercalation, thus leading to the improved specific capacity (Supplementary Fig. 37).

Supplementary Fig. 36 a) Comparison of CV curves of PANI-Zn batteries at 1 mV s^{-1} with $1 \text{ M Zn(PS)}_2+0.2 \text{ TBATS}$ electrolyte in H_2O and DMF. b) Nyquist plot of PANI-Zn batteries and c) the enlarged curves in the high-frequency range. d) Rate performance.

The second read through also finds more issues.

Comments 1. The experimental section lacks details. For example, the authors presented the data of surface zeta-potential and differential capacitance in Fig. 4, but there is virtually no description on how these data were collected.

Response: Thanks for your kinder reminder. The related experimental details are added in page 17 accordingly.

“Zeta Sizer Malvern Nano-ZS was employed to determine the zeta potential of Zn particle in different electrolytes. The samples were prepared by the ultrasonic treating of the Zn powder in 10.0 mL electrolyte solutions to form uniformly distributed

suspensions.”

“Differential capacitance-potential curves were obtained by impedance methods³⁵ with assembled Ti-Zn asymmetric cells. The capacitance can be calculated from the equation: $C=(2\pi fZ_{im})^{-1}$. where C is the capacitance, Z_{im} is the imaginary component of the impedance, and f is the frequency of the ac perturbation.”

Comments 2. The overpotential of 22.6 mV of Zn/Zn symmetrical cell at 1 mA/cm² should not be considered low as the authors claimed. The author can easily compare literature data with their own data.

Response: Thanks for your good suggestion. A complete table is added for overpotential comparison. It can be seen that the low overpotential in the present case is comparable and even better than those results reported recently.

Supplementary Table 1 The voltage hysteresis comparison of Zn-Zn symmetric cell with 1 M Zn(PS)₂+0.2 TBATS electrolyte at 1 mA cm⁻², 1 mAh cm⁻² with recently reports.

Electrode/electrolyte	Overpotential 1 (1st cycle)	Overpotential 2	Ref.
1 M Zn(PS) ₂ +0.2 TBATS	22.6 mV	17.7 mV (200th cycle)	This work
Zn@ZnF ₂	35.7 mV		[1]
Zn(H ₂ PO ₄) ₂ additive	40 mV		[2]
TU/ZnSO ₄		52 mV (27th cycle)	[3]
Cu-Zn@Zn		30 mV	[4]
SPEEK-Zn		50 mV	[5]
La(NO ₃) ₃ additive		<50 mV	[6]
Sn@NHCF-Zn		21 mV	[7]

[1] *Adv Mater* **33**, e2007388 (2021); [2] *Adv Mater* **33**, e2007416 (2021); [3] *Adv. Funct. Mater.* **32**, 2206695 (2022); [4] *Angew. Chem. Int. Ed.* **61**, e202212587 (2022); [5] *Energy Storage Mater.* **49**, 380-389 (2022); [6] *Nat. Commun.* **13**, 3252 (2022); [7] *Sci. Adv.* **8**, eabm5766 (2022).

Comments 3. The dissolution issue of PANI in aqueous solutions needs to be discussed.

Response: Thanks. The related discussion is presented based on the collected data.

The color change of separator into brown is observed possibly due to the decomposition of PANI in ZnSO₄ electrolyte. In comparison with the initial status of PANI (Supplementary Fig. 38a), the nanoflakes corresponding to Zn₄SO₄(OH)₆·H₂O (JCPDS: 39-0690, Supplementary Fig. 39) are formed on the electrode in 1 M ZnSO₄ electrolyte (Supplementary Fig. 38b). The formation of these Zn₄SO₄(OH)₆·H₂O would induce the volume stress, and subsequently result in the structure changes as well as the dissolution of PANI during the cycling stability test. In contrast, no by-product is formed for the electrode in Zn(PS)₂ electrolyte, which would be contributed to the favorable

solvation Zn^{2+} ions with low energy barrier on decoupling the Zn^{2+} cation and PS^- anion. Therefore, the good structure stability of PANI can be achieved in $\text{Zn}(\text{PS})_2$ electrolyte.

Supplementary Fig. 38 SEM images of PANI cathode at a) initial state and photographs of separator after 1000 cycles in b) 1 M ZnSO_4 , c) 1 M $\text{Zn}(\text{PS})_2$ and d) 1 M $\text{Zn}(\text{PS})_2 + 0.2$ TBATS electrolytes.

Supplementary Fig. 39 a) XRD patterns and b) Raman spectra of PANI cathode after 1000 cycles in different electrolytes.

The newly emerged peak at 12.2° after cycling in 1 M ZnSO_4 electrolyte is corresponding to $\text{Zn}_4\text{SO}_4(\text{OH})_6 \cdot \text{H}_2\text{O}$ formed (Supplementary Fig.39a). The peak appears at 968 cm^{-1} is related to the stretching vibration of SO_4^{2-} (Supplementary Fig.39b). In comparison, the PANI cathode demonstrates the good structure stability after 1000 cycles in $\text{Zn}(\text{PS})_2$ electrolyte and no impurity is detected.

Comments 4. It is unclear why the authors selected PANI as the cathode since this is a low-capacity cathode. Have the authors tried high-capacity layered oxides such as V-oxides based materials?

Response: Thanks for your good question.

As a commonly used cathode, the energy storage mechanism of PANI is well-documented [*Nat. Commun.* **14**, 601 (2023); *Proc. Natl. Acad. Sci. U. S. A.* **119**, e2121138119 (2022)]. When used as the positive electrode in the present case, the effect from the PANI electrode can be eliminated. Thus, the understanding on the effect of the new electrolyte on the zinc electrode can be analyzed properly to clarify the underlying mechanism of solvation of zinc ions. Additionally, the fabrication of full battery provides a reasonable example to demonstrate the promising applications of the new electrolyte.

Following your valuable comment, we checked the compatibility with vanadium-oxides (VO) synthesized via a one-pot hydrothermal method. In comparison with the vanadium oxide particles (Fig. R1a, b), the SEM images (Fig. R1c, d) exhibit the uniform flakes which are interconnected each other to form the urchin-like structure. XRD pattern shows that the strongest peak for the (001) plane of V_2O_5 is shifted from 20.3° to 6.4° , suggesting the increasing lattice spacing in comparison with the original V_2O_5 precursor (Fig. R1e), which would facilitate the intercalation of zinc ions. The cyclic voltammogram (CV) curves of VO presents two pairs of asymmetrical redox peaks located at 1.04/0.92 and 0.68/0.53 V in $ZnSO_4$ electrolyte, suggesting the obvious electrochemical polarization with large peak separation of 120 and 150 mV. In contrast, the smaller polarizations of 60 and 120 mV are observed in $Zn(PS)_2$ with/without 0.2 TBATS electrolyte, respectively, showing the improved reversibility (Fig. R2a). The galvanostatic charge/discharge profiles at a current density of 0.5 A g^{-1} show a specific capacity of 381 mAh g^{-1} (Fig. R2b). The VO cathode in 1 M $Zn(PS)_2+0.2$ TBATS electrolyte delivers the high specific capacity of 381, 343, 312, 267, 245, and 236 mAh g^{-1} , respectively at increasing current density from 0.5, 1, 2, 5, 8, to 10 A g^{-1} (Fig. R2c, d). Only 74% capacity retention is achieved after rate examination, revealing the instability in $ZnSO_4$ electrolyte. Especially, the capacity retention is $\sim 82\%$ with 1 M $Zn(PS)_2+0.2$ TBATS electrolyte after 2000 cycles at 5 A g^{-1} which is higher than those in 1 M $ZnSO_4$ and $Zn(PS)_2$ electrolyte (~ 45 and 62%), respectively (Fig. R2e). Impressively, the specific capacity and cycling stability of VO-Zn battery are superior to the vanadium-based aqueous rechargeable ZIBs reported recently (Table R1), demonstrating the superiority of 1 M $Zn(PS)_2+0.2$ TBATS electrolyte.

Fig. R1 SEM images of a, b) V_2O_5 precursor and c, d) VO. e) XRD pattern of V_2O_5 precursor and VO.

Fig. R2 a) Cyclic voltammery curves of VO electrode at 0.2 mV s^{-1} with 1 M ZnSO_4 , 1 M $\text{Zn}(\text{PS})_2$ and 1 M $\text{Zn}(\text{PS})_2+0.2$ TBATS electrolytes. b) Galvanostatic charge/discharge profiles of VO at 0.5 A g^{-1} with 1 M $\text{Zn}(\text{PS})_2+0.2$ TBATS electrolyte. c) Rate performance and d) the corresponding charge/discharge curves of VO at various current densities. e) Cycling performance of VO with different electrolytes at 5 A g^{-1} for 2000 cycles.

Table R1 Electrochemical performance comparisons of VO with other reported V-oxides based batteries.

Cathode	Electrolyte	Specific capacity	Rate performance	Cycling performance	Ref.
VO	1 M Zn(PS) ₂ +0.2 TBATS	381 mAh g ⁻¹ at 0.5 A g ⁻¹	236 mAh g ⁻¹ at 10 A g ⁻¹	82% after 2000 cycles at 5 A g ⁻¹	This work
Mn _{0.15} V ₂ O ₅ · nH ₂ O	1 M Zn(ClO ₄) ₂	299 mAh g ⁻¹ at 0.5 A g ⁻¹	150 mAh g ⁻¹ at 10 A g ⁻¹		[1]
NaV ₃ O ₈ ·1.5 H ₂ O	1 M ZnSO ₄ +1 M Na ₂ SO ₄	375 mAh g ⁻¹ at 0.1 A g ⁻¹	165 mAh g ⁻¹ at 4 A g ⁻¹	82% after 1000 cycles at 4 A g ⁻¹	[2]
V ₂ O ₅ ·nH ₂ O	1 M ZnSO ₄	276 mAh g ⁻¹ at 0.5 A g ⁻¹	104 mAh g ⁻¹ at 5 A g ⁻¹	53% after 400 cycles at 2 A g ⁻¹	[3]
V ₂ O ₅	1 M Zn(CF ₃ SO ₃) ₂ with 25 mM Zn(H ₂ PO ₄) ₂ .	203 mAh g ⁻¹ at 0.1 A g ⁻¹	120 mAh g ⁻¹ at 1 A g ⁻¹	88% after 1000 cycles at 0.8 A g ⁻¹	[4]
PANI-V	2 M ZnSO ₄	283 mAh g ⁻¹ at 1 A g ⁻¹	234 mAh g ⁻¹ at 16 A g ⁻¹	70% after 1000 cycles at 1 A g ⁻¹	[5]
GAVOH	2 M Zn(CF ₃ SO ₃) ₂	381 mAh g ⁻¹ at 0.5 A g ⁻¹	215 mAh g ⁻¹ at 2 A g ⁻¹	83% after 1000 cycles at 1 A g ⁻¹	[6]
V ₂ O ₅	2 M Zn(CF ₃ SO ₃) ₂	347 mAh g ⁻¹ at 0.5 A g ⁻¹	194 mAh g ⁻¹ at 8 A g ⁻¹	74% after 800 cycles at 1 A g ⁻¹	[7]
PAVO	3 M Zn(CF ₃ SO ₃) ₂	295 mAh g ⁻¹ at 0.5 A g ⁻¹	190 mAh g ⁻¹ at 5 A g ⁻¹	90% after 100 cycles at 0.1 A g ⁻¹	[8]
V ₂ O ₃	3 M Zn(CF ₃ SO ₃) ₂	165 mAh g ⁻¹ at 0.5 A g ⁻¹	113 mAh g ⁻¹ at 4 A g ⁻¹	81% after 30000 cycles at 5 A g ⁻¹	[9]
NH ₄ ⁺ -V ₂ O ₅	3 M Zn(CF ₃ SO ₃) ₂	430 mAh g ⁻¹ at 0.1 A g ⁻¹	110 mAh g ⁻¹ at 20 A g ⁻¹	78% after 3600 cycles at 10 A g ⁻¹	[10]

[1] *Adv. Funct. Mater.* **30**, 1907684 (2020); [2] *Nat. Commun.* **9**, 1656 (2018); [3] *ACS Nano* **14**, 11809-11820 (2020); [4] *Adv Mater* **33**, e2007416 (2021); [5] *Adv. Energy Mater.* **10**, 2001852 (2020); [6] *Adv. Funct. Mater.* **33**, 2211412 (2023); [7] *Angew. Chem. Int. Ed.* **61**, e202212587 (2022); [8] *Energy Storage Mater.* **38**, 590-598 (2021); [9] *Nat. Commun.* **12**, 6878 (2021); [10] *Proc. Natl. Acad. Sci. U. S. A.* **120**, e2217208120 (2023).

Comments 5. Fig. S5, y-axis, should the unit be mV?

Response: Thanks for your kinder reminder. The Supplementary Fig. 5 is corrected.

Supplementary Fig. 5 Rate performance of Zn symmetric cells at the current density of 0.5, 1, 2, 5 and 10 mA cm⁻² with the charge-discharge capacity of 0.25, 0.5, 1, 2.5 and 5 mAh cm⁻².

Comments 6. There are many other typos that should be double checked and corrected.

Response: Thanks. We check the whole manuscript carefully.

To Reviewer#3

Reviewer #3 (Remarks to the Author):

The authors have addressed all comments very well, the paper is ready to be accepted.

Thank you very much for your recognition and recommendation of our work.

REVIEWERS' COMMENTS

Reviewer #2 (Remarks to the Author):

The authors have provided more satisfying data to address this reviewer's early comments. The paper is acceptable for publication.

Reviewers' Comments, third round review

Reviewer #2 (Remarks to the Author):

The authors have provided more satisfying data to address this reviewer's early comments. The paper is acceptable for publication.

Thank you very much for your recognition and recommendation of our work.